# Qini Curve Estimation under Clustered Network Interference

## Abstract

Qini curves are a widely used tool for assessing treatment policies under allocation constraints as they visualize the incremental gain of a new treatment policy versus the cost of its implementation. Standard Qini curve estimation assumes no interference between units: that is, that treating one unit does not influence the outcome of any other unit. In many real-life applications such as public policy or marketing, however, the presence of interference is common. Ignoring interference in these scenarios can lead to systematically biased Qini curves that over- or under-estimate a treatment policy's cost-effectiveness. In this paper, we address the problem of Qini curve estimation under clustered network interference, where interfering units form independent clusters. We propose a formal description of the problem setting with a randomized experimental study design where the treatment probabilities are known under which we can account for clustered network interference. Within this framework, we describe three estimation strategies, each suited to different conditions, and provide guidance for selecting the most appropriate approach by highlighting the inherent bias-variance trade-offs. To complement our theoretical analysis, we introduce a marketplace simulator that replicates clustered network interference in a typical e-commerce environment, allowing us to evaluate and compare the proposed strategies in practice. We further demonstrate the applicability of our methods using a semi-synthetic dataset derived from a large-scale real-world marketing experiment.

## 1 Introduction

Understanding treatment effect heterogeneity – the variation in individual responses to the same treatment within a population – is central in shaping individualized treatment policies across various domains, including personalized medicine (Kravitz et al., 2004), uplift modeling in marketing and e-commerce (Goldenberg et al., 2020), and targeted subgroup interventions in public policy (Brand & Davis, 2011). In these scenarios, the same questions recurs: *Who should we treat?* Sometimes, it is sufficient to identify individuals who respond positively to a treatment. However, when treatments involve monetary or practical costs, the challenge is to devise a cost-effective policy that targets those who benefit the most from the treatment while staying within a given budget for treatment allocation.

First introduced by Radcliffe (2007) in the marketing literature, Qini curves have become a widely used method for evaluating the cost-effectiveness of treatment policies. To illustrate how Qini curves are used in practice, we present a potential application from the uplift modeling literature. This application also motivates the setup of the marketplace simulator used later in our simulation studies. Consider a digital marketplace in which customers browse an assortment of products. The outcome of interest is whether a customer purchases a product, and the treatment corresponds to offering a promotion for a given customer–product pair. Each treatment incurs a financial cost, and the goal is to determine whether a treatment policy uses the available promotion budget effectively to improve return on investment, i.e., to generate more purchases.

In this setting, as shown in Figure 1, a Qini curve can be constructed by letting the x-axis represent the expected cost (the financial cost of promotions) and the y-axis the expected incremental gain (the increase in the number of purchased products) relative to if nobody would receive a promotion, as we vary the fraction of would-be treated customers on a test dataset obtained from a randomized experiment on the marketplace. Thus, a Qini curve plots the incremental gain by treating units prioritized by a given treatment rule under

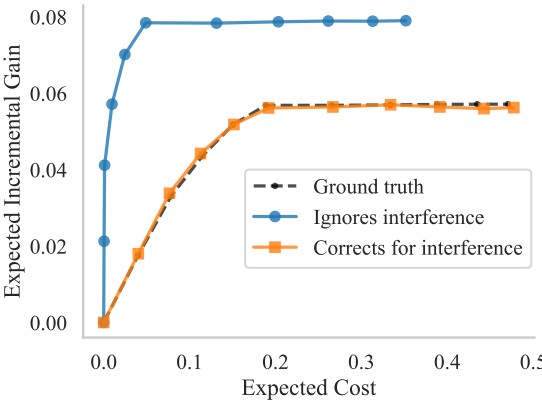

Figure 1: An illustrative simulation study with Qini curve estimation under clustered network interference. The black dashed line represents the true underlying Qini curve, while the solid lines depict two estimation approaches: one based on a traditional method that assumes no interference, and the other representing a proposed strategy in this paper that adjusts for interference using inverse probability weighting. More details on the simulation used to generate this figure can be found in Appendix D.

varying allocation budgets. By comparing the Qini curves of different prioritization rules, practitioners can determine which rule most effectively identifies who responds well to treatment. However, reliable estimation of Qini curves depends on some key assumptions being met, one of which is the Stable Unit Treatment Value Assumption (Rubin, 1980). This assumption implies that there is no treatment interference, meaning that treating one unit has no influence on the outcome of any other unit.

Interference arises in a variety of contexts, from peer effects in social networks (Manski, 2013; Ogburn et al., 2020) to cannibalization effects on marketplace platforms (Holtz et al., 2024). One of the most common settings is so-called *clustered network interference* where interfering units form independent clusters; here units interfere within, rather than between, clusters. While there exists an extensive body of literature on estimating treatment effects under clustered network interference, e.g. Sobel (2006); Hudgens & Halloran (2008), little attention has been given to the problem of estimating Qini curves in this setting. As we demonstrate in Figure 1, traditional methods for estimating Qini curves become significantly biased when interference is present. Since biased Qini curves lead to incorrect assessments of the cost-effectiveness of treatment policies, this is an important yet unaddressed problem. For example, Imai & Li (2023) emphasize the need for methods that evaluate individualized treatment strategies under interference. Motivated by this gap, the central question we aim to answer in this paper is: *How can we account for interference when estimating Qini curves, and how does this adjustment affect decision-making using Qini curves?*

**Contributions** To address our research question, we first formulate the necessary identification conditions for estimating Qini curves under clustered network interference in an experimental study design where the treatment probabilities are known. Next, we describe three separate estimation strategies based on different assumptions of the underlying interference. Our theoretical analysis demonstrates that stronger assumptions on the interference can yield more efficient estimators, though at the potential cost of increased bias when those assumptions are violated. Our main contribution is a systematic study of these bias–variance trade-offs, both theoretically and empirically. The empirical analysis is carried out using a simulated data-generating process that mimics a digital marketplace with different forms of interference, such as cannibalization across vendors, as well as a semi-synthetic dataset derived from a large-scale real-world marketing experiment (Diemert Eustache, Betlei Artem et al., 2018). Finally, based on our findings, we offer practical recommendations for estimating Qini curves in settings with clustered network interference.

## 2 Related works

The task of estimating treatment effects becomes considerably more complex in the presence of interference. Hence, despite early influential works in causal inference such as Rubin (1974), only recently has a substantial body of literature emerged to address interference. One of the most commonly studied settings is clustered network interference, where treatment units form independent clusters (Sobel, 2006; Hudgens & Halloran, 2008; Tchetgen & VanderWeele, 2012), a condition also known as partial interference. Interference also naturally arises in network data, where some units are related to other units by being neighbors in e.g. a social network (Ugander et al., 2013; Eckles et al., 2017). In some cases, an experimental study design can be constructed in a way to detect and reduce bias from interference, for instance, through a two-stage randomization design (Hudgens & Halloran, 2008) or by stratified randomization across different blocks (Bajari et al., 2021). Prior work has tackled specific tasks under interference, such as heterogeneous treatment effect estimation (Zhao et al., 2024) and policy evaluation/learning (Zhang & Imai, 2025). To our knowledge, however, no prior work has considered the problem of estimating Qini curves in the presence of interference.

Evaluating treatment prioritization rules using Qini curves in settings without interference has gained more attention in recent years (Radcliffe, 2007; Rößler & Schoder, 2022; Bokelmann & Lessmann, 2024). The development of estimations procedures with better statistical inference guarantees has enabled the use of Qini curves in this context (Yadlowsky et al., 2024). While none of the above-mentioned works consider interference, Sverdrup et al. (2025) considers the related problem of estimating Qini curves for combinatorial multi-armed treatments.

Although the term Qini curve is widely used in the literature, it is sometimes referred to as the incremental gains curve (Radcliffe & Surry, 2011), the cumulative uplift curve (Surry & Radcliffe, 2011), or simply the uplift curve (Nassif et al., 2013). As noted by Devriendt et al. (2020), however, these terms are used ambiguously and may denote different quantities, most notably the distinction between the Qini curve and the uplift curve. Our definition falls within the class of joint-relative curves in the taxonomy of Devriendt et al. (2020) (see their Table 2), where the Qini curve and uplift curve happen to coincide. We nevertheless expect our framework to extend naturally to the other categories in their taxonomy, as the primary differences typically arise from alternative normalization conventions.

Combinatorial treatment problems and clustered network interference are inherently connected, as assigning treatments within a cluster can be seen as a combinatorial treatment decision. While causal inference methods exist for such settings, see e.g. Dasgupta et al. (2014); Goplerud et al. (2025), they typically assume a fixed treatment dimension – an assumption too restrictive when cluster sizes also may vary. However, the connection between these high-dimensional treatment settings and our setting underscores the challenge of estimating Qini curves under clustered network interference: as cluster size grows the combinatorial space of possible treatments expands exponentially, leading to a corresponding increase in interactions among units within the cluster. To address this challenge, we propose estimation strategies designed to more accurately estimate Qini curves, even as the cluster size differs or grows.

Finally, in our problem setting we assume the interference structure to be partially known, in the sense that we can construct independent clusters that do not interfere with one another. In practice, this requires forming clusters using domain expertise. There is also work that addresses certain cases in which the interference structure is unknown; see, for example, (Bhattacharya et al., 2020) and (Zhang et al., 2023). We leave this setting outside the scope of the present manuscript.

## 3 Data structure & assumptions

**Notation** We have clusters $i = 1, \dots, N$ and each cluster contains the units $j = 1, \dots, M_i$. A unit can be referred to by the tuple $(i, j)$. For each cluster $i$, we observe pre-treatment covariates $X_i$ in $\mathcal{X} \subseteq \mathbb{R}^{d_x}$. For each unit, we observe pre-treatment covariates $Z_{ij}$ in $\mathcal{Z} \subseteq \mathbb{R}^{d_z}$, a binary treatment $W_{ij} \in \{0, 1\}$, and an outcome of interest $Y_{ij}$ in $\mathcal{Y} \subseteq \mathbb{R}$. The outcome may be binary or continuous. In addition, we also observe a non-negative cost of treatment $C_{ij}$ in $\mathcal{C} \subseteq [0, \infty)$. The cost $C_{ij}$ depends on both the treatment and outcome, and specifically we assume there to be no cost $C_{ij} = 0$ when no treatment is given $W_{ij} = 0$. We consider

cluster-level outcomes and costs which we define as $\tilde{Y}_i = \sum_{j=1}^{M_i} Y_{ij}$ and $\tilde{C}_i = \sum_{j=1}^{M_i} C_{ij}$. We also define the cluster-level treatment which is a binary vector $\mathbf{W}_i = [W_{i1}, W_{i2}, \ldots, W_{iM_i}] \in \{0,1\}^{M_i}$. At last, we assume that all observations are drawn from a distribution $P \in \mathcal{P}$, with respect to which all expectations in the paper are taken and its probability density is represented as $f(\cdot)$. Random variables are denoted by capital letters, while their instantiated values use lowercase. Bold letters denote vectors.

**Clustered network interference**   In the setting of clustered network interference, we assume observations can be divided in independent clusters, with the cluster structure known a priori. Treating one unit belonging to cluster $i$ may influence the outcomes of other units from that same cluster. However, treating units from a different cluster $i'$ will not influence the outcomes of the units in cluster $i$. To define causal effects in this setting, we posit potential (counterfactual) outcomes $Y_{ij}(\mathbf{w})$ corresponding to the outcomes we would observe for an unit $(i,j)$ if the treatment vector $\mathbf{W}_i$ would be set to $\mathbf{w}$ (Tchetgen & VanderWeele, 2012). Analogously, we define the counterfactual cost $C_{ij}(\mathbf{w})$ if $\mathbf{W}_i$ would be set to $\mathbf{w}$.

**Study design**   Throughout this paper, we consider an experimental study design where the unit-level treatments $W_{ij}$ are independently and randomly assigned. The treatment probability is determined by $e_w(x) = \Pr(W = w \mid X = x)$ which is known and the same for all units within a cluster. Following standard convention, we refer to this probability as the propensity score (Rosenbaum & Rubin, 1983). We assume the following conditions are fulfilled by our experimental study design.

**Assumption 1.** *Consistency: if $\mathbf{W}_i = \mathbf{w}$ then $Y_{ij}(\mathbf{W}_i) = Y_{ij}$ and $C_{ij}(\mathbf{w}) = C_{ij}$, for all units $(i,j)$ and treatments $\mathbf{w} \in \{0,1\}^{M_i}$. Conditional exchangeability: for each treatment $\mathbf{w} \in \{0,1\}^{M_i}$, $(Y_{ij}(\mathbf{w}), C_{ij}(\mathbf{w})) \perp\!\!\!\perp \mathbf{W}_i \mid X_i$. Positivity: for each treatment $\mathbf{w} \in \{0,1\}^{M_i}$, if $f(x) \neq 0$ then $\Pr(\mathbf{W}_i = \mathbf{w} \mid X_i = x) > 0$.*

*Consistency* is met when the intervention is unambiguously defined, meaning that no undisclosed variants of the treatment exist. *Conditional exchangeability* corresponds to assuming no unmeasured confounding; specifically, the characteristics captured by cluster-level covariates are sufficient to control for any confounding between treatment assignment and outcome/cost. *Positivity* necessitates that all clusters have a non-zero probability of receiving any of combination of available treatments among its units. In the context of our experimental study design, we emphasize that conditional exchangeability and positivity can be guaranteed by (conditional) randomization.

Note that, compared to the standard causal inference setting without interference, the conditions in Assumption 1 are essentially the same, except that the treatment variable is multivariate in our case; without interference, the treatment would typically be one-dimensional.

## 4   Assessing treatment policies using Qini curves under clustered network interference

We are interested in assessing treatment policies based on some treatment prioritization rule $S : \mathcal{X} \times \mathcal{Z} \to \mathbb{R}$ that attempts to rank all units across the clusters based on who responds best to the treatment. A larger $S(X_i, Z_{ij})$ should here be interpreted as that units with covariates $(X_i, Z_{ij})$ are expected to have a larger treatment effect. Given a treatment prioritization rule $S$ and a fixed treatment threshold $R \in \mathbb{R}$, we will evaluate decisions by treatment policies defined as

$$\pi_{S,R}(x,z) = \begin{cases} 1, & S(x,z) \geq R \\ 0, & S(x,z) < R \end{cases}. \tag{1}$$

Importantly, throughout this paper, we will assume that all treatment prioritization rules are derived independently of the data we will use to evaluate them on. For instance, $S$ could be the estimated model of the conditional average treatment effect, see e.g. Künzel et al. (2019), trained on a separate dataset or a formal prioritization rule developed by experts using domain knowledge. To simplify notation, we will omit the subscripts in $\pi_{S,R}$ when possible and simply write $\pi$ to denote a policy.

For clarity and consistency with prior work on Qini curve estimation, we present scoring rules that depend only on unit- and cluster-level covariates, ignoring interference effects. In principle, however, scoring rules

could incorporate treatments and covariates of other units within the same cluster. Notably, the identifiability results at the end of this section remain valid under such generalized scoring rules, though policies based on them may be difficult to implement in practice.

## 4.1 Definition of the Qini curve

A common approach to assess how well a treatment prioritization rule identifies those who best respond to a treatment is by using Qini curves (Radcliffe, 2007; Sverdrup et al., 2025). We denote the decisions made by the policy $\pi$ on an evaluation dataset as $\pi_{ij} = \pi(X_i, Z_{ij}) \in \{0, 1\}$ for unit $(i, j)$ and $\boldsymbol{\pi}_i = [\pi(X_i, Z_{i1}), \ldots, \pi(X_i, Z_{iM_i})] \in \{0, 1\}^{M_i}$ for the collective treatment decision on cluster $i$. Then, we first define the policy value in terms of average cluster-level outcomes under decisions made by the policy $\pi$ as

$$V(\pi) = \mathbb{E}\left[\sum_{j=1}^{M_i} Y_{ij}(\boldsymbol{\pi}_i)\right], \tag{2}$$

and the policy cost of applying $\pi$ as

$$C(\pi) = \mathbb{E}\left[\sum_{j=1}^{M_i} C_{ij}(\boldsymbol{\pi}_i)\right]. \tag{3}$$

We now define the Qini curve for a treatment prioritization rule $S$ as a function of a treatment threshold $R$:

$$Q_S(R) = V(\pi_{S,R}) - V(\pi_0), \tag{4}$$

where $\pi_0 \equiv 0$ is a policy that treats no units. To include costs, we define the budget used by the policy as $B_S(R) = C(\pi_{S,R}) - C(\pi_0)$. If untreated units have zero cost, this simplifies to $B_S(R) = C(\pi_{S,R})$.

To construct the Qini curve, we choose $K$ different treatment thresholds $(R_1, \ldots, R_K)$, for example the 10th, 20th, etc. percentiles of the scores produced by $S$. For each $R_k$, we estimate and plot the pair $(B_S(R_k), Q_S(R_k))$; the next section describes how these quantities are estimated.

The above definition is more general than the one by Radcliffe (2007), who assumes uniform cost across all units. In their approach, one plots $Q_S(B)$ on the y-axis against the fraction of treated units on the x-axis. Our definition reduces to this case by setting $B_S(R) = P_{X,Z}(S(X, Z) \geq R)$, which equals the share of treated units. We refer to this as the uniform cost case.

As we will discuss shortly, the estimation strategies we present apply equally to both the uniform and non-uniform cost settings. Therefore, without loss of generality, we focus on the uniform cost case for the remainder of the paper and note any relevant differences when necessary.

To understand how interference introduces challenges in the estimation of Qini curves, consider a scenario where two units from the same cluster fall on opposite sides of the treatment threshold $R$ – one above (indicating it should be treated) and one below (indicating it should not). Normally, these units would be considered independent, but in the presence of interference, spillover effects may occur between them. If these effects are not accounted for, we might over- or underestimate the policy value and cost which also affects the Qini curve estimation.

For the remainder of this paper, we will demonstrate how to address this issue and provide a methodology for estimating Qini curves that appropriately accounts for interference within a clustered network setting. However, before we propose multiple estimation strategies, we also establish a necessary identifiability result that enables the estimation.

## 4.2 Identifiability of policy value and policy cost

To estimate $Q_S(B)$ in (4), $V(\pi)$ and $C(\pi)$ must be identifiable from the observed data. Under a study design that satisfies Assumption 1, this identifiability is guaranteed. More specifically, recall that $\tilde{Y}_i = \sum_{j=1}^{M_i} Y_{ij}$

---

**Algorithm 1** Qini curve estimation

---

**Require:** Dataset $D = \{X_i, \{Z_{ij}, W_{ij}, Y_{ij}, C_{ij}\}_{j=1}^{M_i}\}_{i=1}^{N}$; treatment prioritization rule $S$; number of percentiles $K$; estimators $\widehat{\phi}$ and $\widehat{\psi}$; Boolean flag indicating if cost is uniform
 1: $\widehat{V}_0 \leftarrow \widehat{\phi}(\pi_0 \equiv 0)$
 2: $(\widehat{B}_0, \widehat{Q}_0) \leftarrow (0, 0)$
 3: Let $S_{\text{sorted}}(i)$ be the score for the $i$th unit when sorted by $S$ in descending order
 4: **for** $k = 1$ to $K$ **do**
 5: $\quad i_k \leftarrow \text{round}\left(\frac{k}{K} \cdot |D|\right)$
 6: $\quad R_k \leftarrow S_{\text{sorted}}(i_k)$
 7: $\quad$ **if** cost is uniform **then**
 8: $\quad\quad \widehat{B}_k \leftarrow \frac{k}{K}$
 9: $\quad$ **else**
10: $\quad\quad \widehat{B}_k \leftarrow \widehat{\psi}(\pi_{S,R_k})$
11: $\quad$ **end if**
12: $\quad \widehat{Q}_k \leftarrow \widehat{\phi}(\pi_{S,R_k}) - \widehat{V}_0$
13: **end for**
14: **return** $\{(\widehat{B}_k, \widehat{Q}_k)\}_{k=0}^{K}$

---

and $\tilde{C}_i = \sum_{j=1}^{M_i} C_{ij}$, then we define

$$\phi(\pi) = \mathbb{E}\big[\mathbb{E}\big[\tilde{Y} \mid \mathbf{W} = \boldsymbol{\pi}, X\big]\big] \quad \text{and} \quad \psi(\pi) = \mathbb{E}\big[\mathbb{E}\big[\tilde{C} \mid \mathbf{W} = \boldsymbol{\pi}, X\big]\big] . \tag{5}$$

With these definitions, we obtain the following important identifiability result (see proof in Appendix A.1).

**Lemma 1.** *Under Assumption 1, we have $V(\pi) = \phi(\pi)$ and $C(\pi) = \psi(\pi)$.*

With the established identification results, we can now outline the general procedure for Qini curve estimation in our experimental study design. In practice, to estimate the Qini curve for a treatment prioritization rule $S$, one typically estimates the policy value $\phi(\pi)$ and cost $\psi(\pi)$ over a range of pre-specified thresholds $R$. For the uniform cost case, it is sufficient to only estimate the policy value. We outline the full procedure in Algorithm 1. So far, we assumed ranking according to $S$ leads to no ties, but if there are ties one could add tiebreakers by, e.g., injecting a small amount of random noise. To implement this algorithm we would need estimators for $\phi(\pi)$ and $\psi(\pi)$; for this reason, next we describe three estimation strategies designed for different scenarios.

## 5 Estimation strategies

In this section we consider three strategies for estimating Qini curves under clustered network inference. In particular, we will focus on weighting estimators that require the propensity score, $e_w(x) = \Pr(W = w \mid X = x)$. Since we assumed the propensity score to be known in our design, we avoid the need to estimate any other nuisance models. Doing so avoids the risk of introducing additional biases due to misspecification when having to estimate nuisance models. Although weighting estimators are the main focus of this paper, we also present in Appendix B some preliminary investigations into techniques for augmenting weighted estimators by incorporating nuisance models that predict the outcome.

Throughout this section, due to the similarity of the estimands for the policy value $\phi(\pi)$ and cost $\psi(\pi)$, we only present strategies for estimating the policy value $\phi(\pi)$ which analogously can be applied for estimating the policy cost $\psi(\pi)$ as well.

### 5.1 Cluster-level inverse probability weighting

The simplest estimator for $\phi(\pi)$ in our setting is

$$\widehat{\phi}^{\text{IPW}}(\pi) = \frac{1}{N} \sum_{i=1}^{N} \frac{\mathbf{1}\left(\mathbf{W}_i = \boldsymbol{\pi}_i\right)}{\prod_{j=1}^{M_i} e_{\pi_{ij}}\left(X_i\right)} \tilde{Y}_i \tag{6}$$

where $\mathbf{1}(\cdot)$ denotes the indicator function. This estimator is a natural extension of the traditional inverse probability weighting (IPW) estimator (Robins et al., 1994) to settings with clustered network interference, see e.g. Tchetgen & VanderWeele (2012). For this reason, we will refer to $\widehat{\phi}^{\,IPW}(\pi)$ as the standard IPW estimator. We can show the following important properties of the standard IPW estimator (see proof in Appendix A.2).

**Theorem 1.** *Under Assumption 1, the standard IPW estimator $\widehat{\phi}^{\,IPW}(\pi)$ is an unbiased estimator for the policy value, $\mathbb{E}[\widehat{\phi}^{\,IPW}(\pi)] = V(\pi)$, and its sampling variance can be written as*

$$\mathrm{Var}\left(\widehat{\phi}^{\,IPW}(\pi)\right) = \frac{1}{N^2} \sum_{i=1}^{N} \left\{ \mathbb{E}\left[\omega(X_i)\left[\tilde{Y}_i(\boldsymbol{\pi}_i)\right]^2\right] + \mathrm{Var}\left(\tilde{Y}_i(\boldsymbol{\pi}_i)\right) \right\} \tag{7}$$

$$\omega(X_i) = \left[ \prod_{j=1}^{M_i} \left( \frac{e_1(X_i)e_0(X_i)}{e_{\pi_{ij}}(X_i)^2} + 1 \right) - 1 \right] \tag{8}$$

From the above theorem, we observe that while the standard IPW estimator is unbiased, its efficiency is poor which becomes evident from inspecting its sampling variance. Since $e_1(X_i) \neq 0$ and $e_0(X_i) \neq 0$ due to Assumption 1, the factor $\omega(X_i)$ increases exponentially with the cluster size $M_i$. Consequently, its variance scales exponentially with the cluster size $M_i$ which makes it prohibitively difficult to use the standard IPW estimator for Qini curve estimation in scenarios where the cluster size $M_i$ is large.

For this reason, we explore other weighting estimators that introduce additional conditions on the structure of the underlying interference. It is important to emphasize here that these additional conditions are not required for identification of the policy value $V(\pi)$, but invoked for more efficient estimation. As we will see, in the cases where these additional conditions do not hold, their respective estimators may introduce additional biases. This results in an inherent bias-variance trade-off for estimating Qini curves in the presence of interference.

### 5.2 Interference under a fractional exposure mapping

One strategy to deal with interference is by defining exposure mappings (Aronow & Samii, 2017). An exposure mapping is a function $d_{ij} : \{0,1\}^{M_i} \to \mathcal{D}$ between all possible treatment configurations for unit $(i,j)$ and a representation (or, embedding) of the treatment configurations. In essence, we want to map similar treatment configurations to the same "effective treatment" (Manski, 2013). If the space $\mathcal{D}$ has smaller cardinality than the original space $\{0,1\}^{M_i}$, which has cardinality $2^{M_i}$, we have the possibility for more efficient estimation. Any exposure mapping, however, must fulfill the following condition.

**Assumption 2.** *The potential outcomes of a unit $(i,j)$ can be grouped by $d_{ij}$, meaning that $d_{ij}(\mathbf{w}) = d_{ij}(\mathbf{w}')$ implies $Y_{ij}(\mathbf{w}) = Y_{ij}(\mathbf{w}')$ for all $\mathbf{w}, \mathbf{w}' \in \{0,1\}^{M_i}$.*

Here, we consider Qini curve estimation using one of the most common ways to define an exposure map. Namely, assuming that the potential outcome $Y_{ij}(\mathbf{w})$ for a unit $(i,j)$ is only a function of both its own treatment status $W_{ij}$ and the fraction of treated units within the same cluster (Ugander et al., 2013; Bajari et al., 2021). This corresponds to the exposure mapping $d_{ij}(\mathbf{W}_i) = [W_{ij}, \overline{W}_i]$ where $\overline{W}_i = M_i^{-1} \sum_{j=1}^{M_i} W_{ij}$.

Denoting the fraction of treated in cluster $i$ by policy $\pi$ as $\overline{\pi}_i = M_i^{-1} \sum_{j=1}^{M_i} \pi_{ij}$, we define the fractional IPW estimator as follows:

$$\widehat{\phi}^{\,\mathrm{fracIPW}}(\pi) = \frac{1}{N} \sum_{i=1}^{N} \sum_{j=1}^{M_i} \frac{\mathbf{1}\left(W_{ij} = \pi_{ij}, \overline{W}_i = \overline{\pi}_i\right)}{q_{ij}(\boldsymbol{\pi}_i, X_i)} Y_{ij} \tag{9}$$

where $q_{ij}(\boldsymbol{\pi}_i, X_i) = \mathrm{Pr}(W_{ij} = \pi_{ij}, \overline{W}_i = \overline{\pi}_i \mid X_i)$. As we show in Appendix A.3, the probability $q_{ij}(\boldsymbol{\pi}_i, X_i)$ can be expressed in terms of the known propensity score. We can now show that the fractional IPW estimator is unbiased in the setting that the fractional exposure mapping is correctly specified (see proof in Appendix A.4).

**Theorem 2.** *Under Assumptions 1 and 2, we have that the fractional IPW estimator $\widehat{\phi}^{\,fracIPW}(\pi)$ is an unbiased estimator for the policy value, $\mathbb{E}\left[\widehat{\phi}^{\,fracIPW}(\pi)\right] = V(\pi)$.*

Notably, the fractional IPW estimator requires additional assumptions compared to the standard IPW estimator to be unbiased, but its variance scales more favorably with the cluster size $M_i$ because the fractional exposure mapping reduces the cardinality of the treatment space from $2^{M_i}$ to $2(M_i + 1)$ per cluster. This reduction makes estimation more feasible in settings with large clusters. However, this does not imply that the variance of $\widehat{\phi}^{\text{fracIPW}}$ grows linearly with $M_i$. The estimator remains inversely proportional to the probability $q_{ij}(\boldsymbol{\pi}_i, X_i)$. As $M_i$ increases, the number of possible treatment fractions grows, making it less likely to observe a specific fraction. Consequently, $q_{ij}(\boldsymbol{\pi}_i, X_i)$ approaches zero as $M_i$ increases, which amplifies the variance, though at a slower rate than the standard IPW estimator.

### 5.3 Interference under $\beta$-additive model

Next, we consider another strategy that can reduce variance compared to the standard IPW estimator. Specifically, we use the following polynomial model to describe the interference, as proposed by Zhang & Imai (2025).

**Assumption 3.** *The potential outcome model satisfies $\mathbb{E}[Y_{ij}(\mathbf{W}_i) \mid X_i] = \mathbf{g}_j(X_i)^\top \gamma(\mathbf{W}_i)$ where $\mathbf{g}_j(X_i) = [g_j^{(0)}, \dots, g_j^{(m)}]^\top$ is an unknown vector of functions $g_j^{(\cdot)} : \mathcal{X} \to \mathbb{R}$ that may vary across units in the same cluster. Furthermore, we have the augmented treatment vector $\gamma(\mathbf{W}_i) = \left[1, \mathbf{W}_i, \mathbf{W}_i^{(2)}, \dots, \mathbf{W}_i^{(\beta)}\right]^\top$ with $\mathbf{W}_i^{(k)} = \left\{ \prod_{m=1}^k W_{ij_m} \mid j_1 < \dots < j_k \right\}$ that contains interactions up to the order of $\beta$ between treatment of different units in the same cluster. Here, $\beta$ is upper bounded by the largest possible $M_i$.*

This assumption states that each unit's conditional mean potential outcome is a linear function of the augmented treatment vector $\gamma(\mathbf{W}_i)$, which includes interaction terms up to order $\beta$ between treatments within the same cluster. We will refer to the above assumption as the $\beta$-additive assumption.

Denoting $\mathcal{I}_i^\beta$ as the power set of $\{1, \dots, M_i\}$ with cardinality at most $\beta$, we define the $\beta$-additive IPW estimator

$$\widehat{\phi}^{\,\beta\text{-IPW}}(\pi; \beta) = \frac{1}{N} \sum_{i=1}^N \left[ \sum_{\mathcal{U} \in \mathcal{I}_i^\beta} \prod_{j \in \mathcal{U}} \left( \frac{\mathbf{1}(W_{ij} = \pi_{ij})}{e_{\pi_{ij}}(X_i)} - 1 \right) \right] \tilde{Y}_i \;. \tag{10}$$

To use the estimator $\widehat{\phi}^{\,\beta\text{-IPW}}(\pi; \beta)$, we must specify $\beta$. When this parameter is chosen to satisfy Assumption 3 alongside Assumption 1, Zhang & Imai (2025) proved the following result.

**Theorem 3.** *Under Assumptions 1 and 3, we have that the $\beta$-IPW estimator $\widehat{\phi}^{\,\beta\text{-IPW}}(\pi)$ is an unbiased estimator for the policy value, $\mathbb{E}\left[\widehat{\phi}^{\,\beta\text{-IPW}}(\pi; \beta)\right] = V(\pi)$.*

The choice of $\beta$ dictates the strength of the $\beta$-additive assumption, which becomes less restrictive as $\beta$ increases. Setting $\beta = \max_i M_i$ imposes no additional constraints on the interference structure since, in this case, we have that $\widehat{\phi}^{\,\beta\text{-IPW}} = \widehat{\phi}^{\,\text{IPW}}$ (Zhang & Imai, 2025). Thus, from a practical point of view, the greatest variance reduction can be achieved by using a smaller $\beta$.

To highlight the best variance reduction we can possibly achieve with the $\beta$-IPW estimator, we consider the special case of $\widehat{\phi}^{\text{addIPW}}(\pi) = \widehat{\phi}^{\,\beta\text{-IPW}}(\pi; \beta = 1)$, which we refer to as the additive IPW estimator because there are no interactions between multiple treatments within the same cluster. Here, we can establish the following claim (see Appendix A.5 for proof).

**Lemma 2.** *We have that $\mathrm{Var}\left(\widehat{\phi}^{addIPW}\right) \propto \max_i M_i^2$.*

Thus, for the additive IPW estimator, we observe that its variance will scale quadratically with the cluster size. This is a notable improvement over the exponential scaling of the standard IPW estimator.

# 6 Experiments

We aim to evaluate the performance of our proposed strategies for estimating Qini curves under clustered network interference. To balance realistic structures of interference with the benefit of having a ground-truth in synthetic data, we designed a simulator that mimics an e-commerce marketplace where interference arises through cannibalization among product items sold by different vendors. Notably, we implemented various interference structures in which Assumptions 2 and 3 are also violated. We present our simulator as a framework that can be reused for future research on the topic of interference. [1]

In our experiments, we compare five strategies for estimating Qini curves. First, we implement a strategy that ignores all interference, which we refer to as the naive estimator (more details are found in Appendix C). Next, we implement the estimators discussed in this paper: the standard IPW estimator, the fractional IPW estimator and the $\beta$-IPW estimator with $\beta = 1$ (additive IPW) or $\beta = 2$.

**Evaluation criteria** Our experiments focus on two key aspects of using Qini curves for decision-making. The first aspect is calibration: how accurately the estimates $\{\widehat{Q}_k\}_{k=0}^K$ reflect the ground truth values $\{Q_k\}_{k=0}^K$. Depending on the experiment, we assess this using bias $K^{-1}\sum_{k=1}^K \mathbb{E}[\widehat{Q}_k - Q_k]$, variance $K^{-1}\sum_{k=1}^K \text{Var}(\widehat{Q}_k)$, and mean squared error $K^{-1}\sum_{k=1}^K \mathbb{E}[(\widehat{Q}_k - Q_k)^2]$. The second aspect is discrimination: the ability to determine which policy is better. For this, we rank policies based on the estimated area under the Qini curve (higher is better) and use Kendall rank correlation to measure how well each estimator ranks policies compared to the ground truth ranking. By default, since our focus lies on evaluating treatment policies rather than the policies themselves, we implement a simple baseline policy with access to the underlying data-generating process. We then progressively degrade its performance by adding increasing levels of noise to its scoring rule. More details on this policy are provided in Appendix D.1. To simplify evaluation, we perform experiments in the uniform cost case.

## 6.1 Simulating an e-commerce marketplace with clustered network interference

In this section, we describe a data-generating process where clusters correspond to potential buyers searching for some item, while treatment units are the items shown. In this marketplace, the treatment of an item $W_{ij}$ could correspond to e.g. discounts or promotions, and the outcome $Y_{ij}$ is whether the item $(i, j)$ was purchased by the buyer $i$. Treatment effects manifest as an incremental change in the probability of a purchase to occur due to the treatment. Each buyer can make at most one purchase, causing cannibalization as treatments may shift purchases between items rather than increasing total purchases.

To construct the dataset, we first sample the covariates $(X_i, Z_{ij})$ and treatment $W_{ij}$. Next, to introduce heterogeneous treatment effects, we compute an item attractiveness score matrix $\mathbf{A}$, where each element $A_{ij} \in [0, 1]$ represents buyer $i$'s interest in purchasing item $(i, j)$. The elements in $\mathbf{A}$ depend on the covariates and assigned treatment. Details on this sampling and computation are provided in Appendix D.1. For simplicity, we assume all buyers observe the same number of items, denoted by $M$.

Next, before sampling the outcome $Y_{ij}$, which indicates whether item $(i, j)$ is purchased, we first determine whether buyer $i$ makes any purchase at all, denoted with the binary variable $\tilde{Y}_i$. Conditional on a purchase occurring, we then sample which specific item the buyer purchases. We sample $\tilde{Y}_i$ according to the Bernoulli probability $\eta(\mathbf{A}_i) = P(\tilde{Y}_i = 1 \mid \mathbf{A}_i)$. The structure of the interference in this dataset is largely determined by the choice of $\eta$.

We consider three alternatives for $\eta$. The simplest is $\eta_{\max}(\mathbf{A}_i) = \max_j A_{ij}$ where only the most attractive item contributes the probability of a purchase by buyer $i$. We refer to $\eta_{\max}$ as the max function. Next, we consider the product function $\eta_{\text{product}}(\mathbf{A}_i) = 1 - \prod_{j=1}^M (1 - A_{ij})$. This function assumes that items contribute independently to a purchase such that $P(\tilde{Y}_i = 1) = 1 - P(\tilde{Y}_i = 0) = 1 - \prod_{j=1}^M P(Y_{ij} = 0)$. Lastly, the third function is inspired by position bias, commonly found in ranking systems used in e-commerce platforms (Joachims et al., 2005). We refer to this as the exponential decay function, defined as $\eta_{\text{exp-decay}}(\mathbf{A}_i) = \sum_{j=1}^M (\frac{1}{2})^{\text{rank}(A_{ij})} A_{ij}$, where $\text{rank}(\cdot)$ returns the rank of the attractiveness scores for buyer $i$ in descending

---

[1] We provide all code for reproducing our simulations in the supplementary materials during the review process.

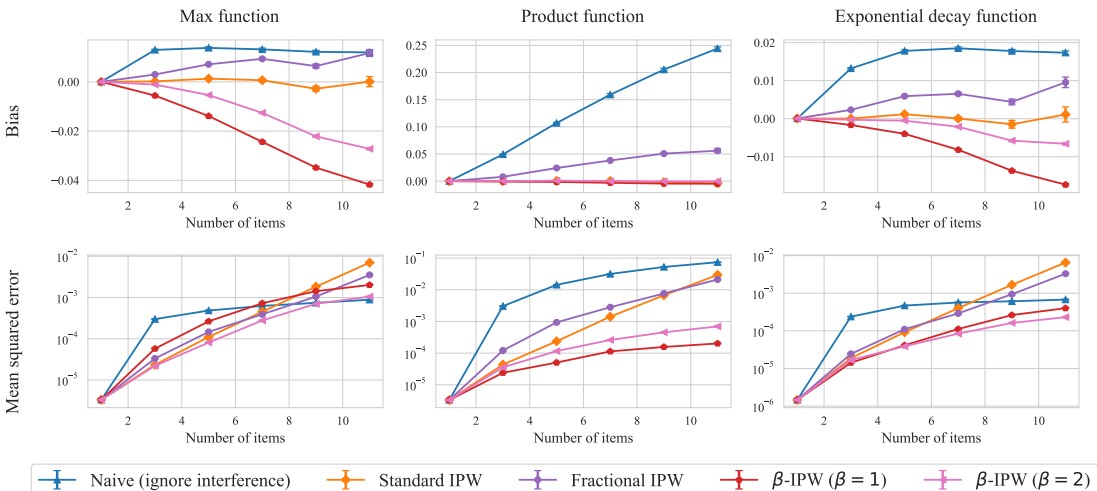

Figure 2: Comparison of bias and mean squared error for each strategy under different interference structures. We let $N = 100.000$ and $M = 11$. Averages and standard errors are reported over 150 repetitions.

order. Each successive item contributes half as much as the preceding one to the probability of a purchase by buyer $i$.

For the final step, conditional on that we sample $\tilde{Y}_i = 1$, we determine which item $(i, j)$ is purchased by sampling according to the probabilities given by the softmax function $P(Y_{ij} = 1 \mid \tilde{Y}_i = 1, \mathbf{A}_i) = e^{A_{ij}/\lambda} / \sum_{j=1}^{M} e^{A_{ij}/\lambda}$ where we set the temperature parameter $\lambda = 0.1$.

### 6.2 How does interference affect the estimation error?

In the first experiment, we evaluated all estimation strategies under different interference structures by varying $\eta$ as described in the previous subsection. In addition, we fixed the number of buyers (i.e., clusters) while varying the number of items $M$ (i.e., units), since spillover effects due to interference are largely expected to depend on cluster size; when the cluster size is one, there is no interference. We compared the bias and mean squared error (MSE) of each estimation strategy, as shown in Figure 2.

Starting with the bias, we observed that the naive estimator is significantly biased for all interference structures. The bias, which increased with number of items, was generally smaller for the fractional IPW and $\beta$-IPW estimators with $\beta = 2$. Meanwhile, the standard IPW estimator appeared unbiased in all cases. In all cases, the $\beta$-IPW estimator with $\beta = 2$ is observed to have lower bias than the variant with $\beta = 1$ and, in some cases, for larger number of items (e.g., more than 10) the variant with $\beta = 1$ had a similar or larger absolute bias as the naive estimator.

In terms of MSE, the standard IPW and fractional IPW estimators performed worst, with MSE increasing exponentially with cluster size. The $\beta$-IPW estimators performed the best; in particular, for the product function, $\beta = 1$ yielded the best MSE. The naive estimator performed the worst for small item counts but its increase in MSE appeared to slow down for larger number of items.

Finally, by visually examining the average Qini curve for a fixed number of items $M = 11$ with the exponential decay function, as in Figure 3, we can obtain a more qualitative assessment of the bias of each estimation strategy. Starting from the left, we observe that the naive estimator vastly overestimates the Qini curve; the standard IPW is centered around the true curve; fractional IPW also overestimates, but to a lesser degree than the naive; and the $\beta$-IPW estimators underestimate the Qini curves, where $\beta = 2$ has less bias than $\beta = 1$ as expected. We further examined the average difference in the AUC of the estimated Qini curves and the ground-truth ones across all interference structures, and we observe similar trends for the other interference structure (max and product); the full results are provided in Appendix D.5.

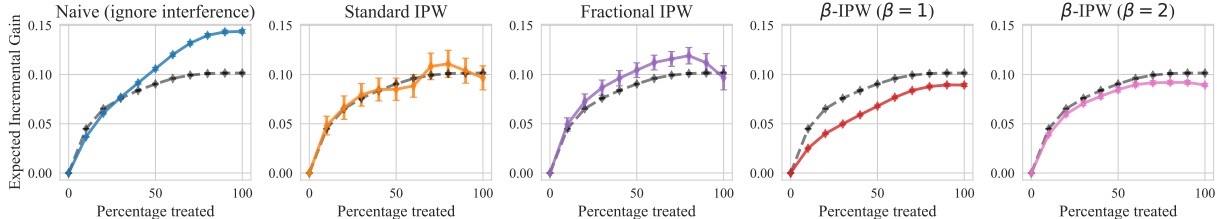

Figure 3: Qini curves of each estimation strategy with $N = 100.000$ and $M = 11$ using with the exponential decay function $\eta_{\text{exp-decay}}$. The average Qini curve and standard error are reported over 150 repetitions. The dashed black line corresponds to the true underlying Qini curve.

Table 1: Comparison of ability to rank policies by each estimation strategy. We used $N = 20.000$ number of buyers (i.e., clusters) and $K = 11$ items (i.e., units) per buyer. We report the average Kendall rank correlation with respect to the ground truth ranking over 150 repetitions. Higher is better, where 1 corresponds to a perfect rank correlation.

| Estimation strategy | Max | Product | Exponential decay |
|---|---|---|---|
| Naive (ignore interference) | 1.000 | 0.808 | 1.000 |
| Standard IPW | 0.928 | 0.845 | 0.945 |
| Fractional IPW | 0.991 | 0.806 | 0.995 |
| $\beta$-IPW ($\beta = 1$) | 1.000 | 1.000 | 1.000 |
| $\beta$-IPW ($\beta = 2$) | 1.000 | 0.995 | 1.000 |

### 6.3   Which estimation strategy is most efficient?

In the next experiment, we evaluated the efficiency of each estimation strategy by reporting the variance as we varied the number the buyers $N$ (i.e., clusters) or items $M$ (i.e., units). While varying one, the other was kept fixed to $N = 20.000$ and $K = 11$. We present results only using $\eta_{\text{exp-decay}}$ as we observed no difference when changing this function. The results are shown in Figure 4 in the appendix, and we summarize our main findings below.

The results indicate that the most efficient estimators, ranked from lowest to highest variance, are: naive, $\beta$-IPW ($\beta = 1$), $\beta$-IPW ($\beta = 2$), fractional IPW, and standard IPW. While the naive has the lowest variance, we recall that it also has the largest bias since it fails to take into account the interference. As expected, the variance of all estimators improved with more buyers (i.e., clusters) at a similar rate. Meanwhile, the variance of the standard and fractional IPW estimators increased exponentially with the number of items (i.e., cluster size), whereas the others scaled sub-exponentially.

### 6.4   How well can we rank policies under clustered network interference?

In the final experiment, we evaluated each estimation strategy's ability to rank policies based on the estimated area under the Qini curve. To do so, we degraded the baseline treatment policy used for evaluation by adding progressively larger noise to its treatment prioritization rule, generating seven policies with decreasing performance. This experiment was repeated 150 times with $N = 20.000$ and $K = 11$ for each $\eta$, and we report the average Kendall rank correlation coefficient between each strategy's ranking and the ground truth.

The results, shown in Table 1, indicate that $\beta$-IPW with $\beta = 1$ performed best overall, achieving perfect rankings in all cases (rank correlation $= 1$). The other estimators achieved similar or only slightly lower correlations, except for $\eta_{\text{product}}$, where the naive estimator, standard IPW, and fractional IPW had correlations between 0.80 and 0.85.

# 7 Case study: Large-scale CRITEO uplift dataset

We use the public CRITEO uplift dataset from a large randomized advertising experiment (Diemert Eustache, Betlei Artem et al., 2018) to evaluate our methods in a realistic setting. The anonymized data contain approximately 14 million users with 12 pre-treatment covariates, a binary treatment indicating whether an advertisement was shown, and two binary outcomes (visit and conversion within a two-week test period). In this case study, we let the conversion be our outcome of interest. The overall percentage of treated users is 84.6% in the dataset.

Because the data from the original experiment is assumed to satisfy the assumption of no interference, we create a semi-synthetic version of the data in which we inject clustered network interference. To do so, we first estimate the baseline outcome probability $\hat{p}_m = P(Y = 1 \mid X_m, A_m)$ using a gradient boosting model with two-fold cross-fitting for every user $m$. Next, users are randomly assigned to clusters of sizes ranging from 2 to 5, which we denote by a cluster index $c(m)$. For each user $m$ we then compute $u_m = \text{logit}(\hat{p}_m) + \tau_{\text{spill}} \cdot g_m$, where $g_m = \log\left(1 + \frac{1}{|c(m)|-1} \sum_{n \in c(m),\, n \neq m} A_n\right)$. The parameter $\tau_{\text{spill}}$ controls the strength of the spillover effect, which we set to 0.25. Finally, we sample a new outcome as $Y_m \sim \text{Bern}\left(\frac{1}{1+e^{-u_m}}\right)$.

This dataset construction induces clustered interference into the original dataset such that each unit's outcome probability is also affected by the fraction of treated units in its own cluster. Because the spillover effects between units depend solely on this fraction, Assumption 2 is satisfied. Our goal is however not to model a realistic network, but to evaluate the estimators under controlled interference while preserving the real data distribution and realistic treatment policies. In this case study, we also focus on comparing the behavior of the estimators rather than finding a winner among the different estimators.

## 7.1 Results

We split the data into training and test sets and construct two policies. Policy A makes decisions using the predictions from a T-learner with a linear logistic regression fitted for each treatment arm, while Policy B uses the same model with added Gaussian noise to degrade its predictions; in this way we would expect Policy A to perform better than policy B. We evaluate both policies on the test set using the AUC of the Qini curve obtained by each estimator. This procedure is repeated 50 times, and we report the mean AUC and its standard error.

As shown in Table 2, the naive estimator differs markedly from standard and fractional IPW, which are both expected to be unbiased in this setting. This indicates that the naive estimator is biased. In contrast, the $\beta$-IPW estimators are close to standard and fractional IPW, suggesting only small bias, while exhibiting substantially lower variance as reflected in their standard errors – nearly matching the variance of the naive estimator. We thus observe patterns consistent with the simulation study. First, $\beta$-IPW achieves the largest variance reduction among our proposed estimators, which would be important for model selection, and fractional IPW appears unbiased when the interference depends only on the fraction treated within each cluster. Finally, even though the naive estimator also has low variance, it is poorly calibrated and yields a substantially biased Qini curve.

# 8 Discussion

Our findings indicate that, while clustered network interference can cause severe bias in Qini curve estimates when the interference is ignored, it is possible to get accurate estimates using different estimation strategies that take this interference into account. However, the best estimation strategy will depend on several application-specific factors, including cluster size, the number of observations, and prior beliefs about interference and the intended use of the Qini curve.

For small cluster sizes (e.g., fewer than 5), the choice of estimation strategy had a limited impact on estimation error, as IPW, fractional IPW, and $\beta$-IPW all performed comparably in terms of bias and mean squared error. For larger cluster sizes, however, we observed a trade-off between using an unbiased, high-variance estimator and a possibly biased, low-variance estimator. For unbiased estimation, the standard IPW estimator is

Table 2: Results obtained by applying the estimators to a semi-synthetic dataset constructed from a large-scale randomized advertising experiment. We report the average AUC of the Qini curves for each estimator and its standard error over 50 runs with different train/test splits.

| Estimator | Policy A | | Policy B | |
|---|---|---|---|---|
| | Mean | SE | Mean | SE |
| Naive | 1228.9 | 75.8 | 915.3 | 28.0 |
| Standard IPW | 2894.7 | 341.7 | 2375.5 | 311.2 |
| Fractional IPW | 2825.4 | 331.4 | 2391.5 | 289.7 |
| $\beta$-IPW ($\beta = 1$) | 2991.5 | 101.8 | 2486.2 | 40.6 |
| $\beta$-IPW ($\beta = 2$) | 2649.5 | 170.2 | 2234.5 | 130.3 |

preferred as it relies only on the weakest conditions regarding the interference, though it requires a large number of observations to be reliable. When the number of observations is limited, low-variance unbiased estimation might still be feasible if strong domain expertise about the interference structure can justify the use of either the fractional IPW or the $\beta$-IPW estimator. However, if some bias is tolerated, then our results suggest that $\beta$-IPW is a strong choice, as it has the lowest variance among strategies that account for interference.

The acceptable level of bias depends on the decision-making context. For model selection – i.e., discriminating between good and bad policies – we observed the $\beta$-IPW estimator performing best. Interestingly, the naive estimator that ignores interference also ranked policies correctly in many cases, suggesting that bias due to interference may have a limited effect on this type of decision-making. However, if the goal is to determine a suitable threshold for a treatment prioritization rule, a well-calibrated Qini curve becomes more critical, making an estimator with low bias preferable.

**Additional considerations when using $\beta$-IPW estimator** Since Assumptions 2 and 3 are untestable, it is difficult to choose between the fractional and $\beta$-IPW estimators without domain expertise. However, for the $\beta$-IPW in particular, an alternative could be to adopt a data-adaptive strategy to find an estimator optimal with respect to the MSE. We can leverage the fact that larger values of $\beta$ make the $\beta$-additivity assumption less restrictive. Recall that as $\beta$ increases, the $\beta$-IPW estimator converges to the standard IPW estimator. This relationship can be exploited to estimate the bias for a given $\beta$ by comparing the $\beta$-IPW estimate to that of the unbiased standard IPW estimator. Combining this bias estimate with an estimate of the sampling variance of the $\beta$-IPW estimator obtained via bootstrapping allows us to compute an MSE estimate for each $\beta$, from which we can select the value that minimizes the estimated MSE. A potential limitation of this approach, however, is that the MSE estimate itself may have high variance, owing to the uncertainty propagated from the standard IPW estimator used as a reference for estimating the bias. Despite this practical challenge, we view this approach as a promising direction for future work toward developing more data-adaptive methods for choosing among the estimators proposed here.

**Incorporating cost into Qini curves** We would like to conclude with some remarks on the potential challenges of defining cost when working with Qini curves. In some applications it is relatively straightforward to specify a cost, such as the financial cost of offering a promotion in the marketplace example we have talked about before. In other settings, however, defining an appropriate cost may be considerably more challenging. In medical applications, for example, the potential side effects of a vaccine can be difficult to quantify. Furthermore, our notion of cost may be counterfactual, in the sense that it can depend on the realized outcome. For instance, in the promotion setting a cost is incurred only if the customer ultimately makes a purchase. This view is closely related to prior work on counterfactual harm (Richens et al., 2022). But when the cost is treated as a counterfactual, the same conditions typically required for identification of quantities involving potential outcomes also need to hold for the cost; these conditions were already outlined in Assumption 1.

## 9    Conclusion

To summarize, we have introduced a framework for estimating Qini curves in experimental study designs with clustered network interference, along with multiple estimation strategies. Our results demonstrate that properly accounting for interference leads to more accurate Qini curve estimation, though the best estimation strategy depends on the specific context. While these methods are not a universal solution for all types of interference and should be applied with care, especially in high-stakes settings, we provide practical recommendations based on both theoretical insights and empirical evidence from the simulated and semi-synthetic data. These guidelines are intended to help practitioners more effectively assess the cost-effectiveness of treatment policies in complex settings where interference is present.

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

# A Proofs and derivations

## A.1 Proof of Lemma 1

*Proof.* Under Assumption 1, we can show that $\mathbb{E}[\phi(\pi)] = V(\pi)$ by rewriting the expectation as follows,

$$
\phi(\pi) = \mathbb{E}\left[\mathbb{E}[\tilde{Y} \mid \mathbf{W} = \boldsymbol{\pi}, X]\right]
$$

$$
= \mathbb{E}\left[\mathbb{E}\left[\sum_{j=1}^{M_i} Y_{ij}(\boldsymbol{\pi}_i) \mid \mathbf{W} = \boldsymbol{\pi}_i, X_i\right]\right]
$$

$$
= \mathbb{E}\left[\mathbb{E}\left[\sum_{j=1}^{M_i} Y_{ij}(\boldsymbol{\pi}_i) \mid X_i\right]\right]
$$

$$
= \mathbb{E}\left[\sum_{j=1}^{M_i} Y_{ij}(\boldsymbol{\pi}_i)\right]
$$

where the second equality from that we defined $\tilde{Y}_i = \sum_{j=1}^{M_i} Y_{ij}$ and then $Y_{ij} = Y_{ij}(\boldsymbol{\pi}_i)$ due to consistency in Assumption 1, and finally the third equality from conditional exchangeability $Y_{ij}(\boldsymbol{\pi}_i) \perp\!\!\!\perp \mathbf{W}_i \mid X_i$ in Assumption 1. We can prove analogously using the same arguments that $\mathbb{E}[\psi(\pi)] = C(\pi)$. $\qquad\square$

## A.2 Proof of Theorem 1

*Proof.* The unbiasedness of the standard IPW estimator $\widehat{\phi}^{\text{IPW}}(\pi)$ follows from the same arguments as deriving the inverse probability weighting estimator in settings with no interference (Hernan & Robins, 2020). We can show that

$$
\mathbb{E}\left[\widehat{\phi}^{\text{IPW}}(\pi)\right] = \mathbb{E}\left[\frac{1}{N}\sum_{i=1}^{N} \frac{\mathbf{1}(\mathbf{W}_i = \boldsymbol{\pi}_i)}{\prod_{j=1}^{M_i} e_{\pi_{ij}}(X_i)}\tilde{Y}_i\right]
$$

$$
= \mathbb{E}\left[\frac{1}{N}\sum_{i=1}^{N} \frac{\mathbf{1}(\mathbf{W}_i = \boldsymbol{\pi}_i)}{\Pr(\mathbf{W}_i = \boldsymbol{\pi}_i \mid X_i)}\tilde{Y}_i\right]
$$

$$
= \mathbb{E}\left[\frac{1}{N}\sum_{i=1}^{N} \mathbb{E}\left[\frac{\mathbf{1}(\mathbf{W}_i = \boldsymbol{\pi}_i)}{\Pr(\mathbf{W}_i = \boldsymbol{\pi}_i \mid X_i)}\tilde{Y}_i \mid X_i\right]\right]
$$

$$
= \mathbb{E}\left[\frac{1}{N}\sum_{i=1}^{N} \mathbb{E}\left[\tilde{Y}_i \mid \mathbf{W}_i = \boldsymbol{\pi}_i, X_i\right]\right]
$$

$$
= \mathbb{E}\left[\phi(\pi)\right]
$$

The second equality follows from the independent treatment assignments where

$$
\Pr(\mathbf{W}_i = \boldsymbol{\pi}_i \mid X_i) = \prod_{j=1}^{M_i} \Pr(W_{ij} = \pi_{ij} \mid X_i) = \prod_{j=1}^{M_i} e_{\pi_{ij}}(X_i),
$$

The unbiasedness of $\widehat{\phi}^{\text{IPW}}(\pi)$ then follows from $\mathbb{E}[\phi(\pi)] = V(\pi)$ under Assumption 1.

Next, we derive the expression for $\text{Var}\left(\widehat{\phi}^{\text{IPW}}(\pi)\right)$. Due to independence of clusters, we first note that

$$
\text{Var}\left(\widehat{\phi}^{\text{IPW}}(\pi)\right) = \text{Var}\left(\frac{1}{N}\sum_{i=1}^{N} \frac{\mathbf{1}(\mathbf{W}_i = \boldsymbol{\pi}_i)}{\prod_{j=1}^{M_i} e_{\pi_{ij}}(X_i)}\tilde{Y}_i\right) = \frac{1}{N^2}\sum_{i=1}^{N} \text{Var}\left(\frac{\mathbf{1}(\mathbf{W}_i = \boldsymbol{\pi}_i)}{\prod_{j=1}^{M_i} e_{\pi_{ij}}(X_i)}\tilde{Y}_i\right).
$$

By using the law of total variance, we can rewrite

$$\text{Var}\left(\frac{\mathbf{1}\left(\mathbf{W}_i = \boldsymbol{\pi}_i\right)}{\prod_{j=1}^{M_i} e_{\pi_{ij}}\left(X_i\right)}\tilde{Y}_i\right) = \mathbb{E}\left[\underbrace{\text{Var}\left(\frac{\mathbf{1}\left(\mathbf{W}_i = \boldsymbol{\pi}_i\right)}{\prod_{j=1}^{M_i} e_{\pi_{ij}}\left(X_i\right)}\tilde{Y}_i \mid \tilde{Y}_i(\boldsymbol{\pi}_i), X_i\right)}_{(a)}\right] + \text{Var}\left(\underbrace{\mathbb{E}\left[\frac{\mathbf{1}\left(\mathbf{W}_i = \boldsymbol{\pi}_i\right)}{\prod_{j=1}^{M_i} e_{\pi_{ij}}\left(X_i\right)}\tilde{Y}_i \mid \tilde{Y}_i(\boldsymbol{\pi}_i), X_i\right]}_{(b)}\right) .$$

Inspecting $(b)$ first, we note that

$$
\begin{aligned}
(b) &= \mathbb{E}\left[\frac{\mathbf{1}\left(\mathbf{W}_i = \boldsymbol{\pi}_i\right)}{\prod_{j=1}^{M_i} e_{\pi_{ij}}\left(X_i\right)}\tilde{Y}_i(\boldsymbol{\pi}_i) \mid \tilde{Y}_i(\boldsymbol{\pi}_i), X_i\right] \\
&= \tilde{Y}_i(\boldsymbol{\pi}_i)\mathbb{E}\left[\frac{\mathbf{1}\left(\mathbf{W}_i = \boldsymbol{\pi}_i\right)}{\prod_{j=1}^{M_i} e_{\pi_{ij}}\left(X_i\right)} \mid \tilde{Y}_i(\boldsymbol{\pi}_i), X_i\right] \\
&= \tilde{Y}_i(\boldsymbol{\pi}_i) .
\end{aligned}
$$

where it follows from conditional exchangeability in Assumption 1 that $\mathbb{E}\left[\frac{\mathbf{1}(\mathbf{W}_i=\boldsymbol{\pi}_i)}{\prod_{j=1}^{M_i} e_{\pi_{ij}}(X_i)} \mid \tilde{Y}_i(\boldsymbol{\pi}_i), X_i\right] = \mathbb{E}\left[\frac{\mathbf{1}(\mathbf{W}_i=\boldsymbol{\pi}_i)}{\prod_{j=1}^{M_i} e_{\pi_{ij}}(X_i)} \mid X_i\right] = 1$.

Similarly, we can show that

$$
\begin{aligned}
(a) &= \text{Var}\left(\frac{\mathbf{1}\left(\mathbf{W}_i = \boldsymbol{\pi}_i\right)}{\prod_{j=1}^{M_i} e_{\pi_{ij}}\left(X_i\right)}\tilde{Y}_i(\boldsymbol{\pi}_i) \mid \tilde{Y}_i(\boldsymbol{\pi}_i), X_i\right) \\
&= \left[\frac{\tilde{Y}_i(\boldsymbol{\pi}_i)}{\prod_{j=1}^{M_i} e_{\pi_{ij}}\left(X_i\right)}\right]^2 \text{Var}\left(\mathbf{1}(\mathbf{W}_i = \boldsymbol{\pi}_i) \mid \tilde{Y}_i(\boldsymbol{\pi}_i), X_i\right) \\
&= \left[\frac{\tilde{Y}_i(\boldsymbol{\pi}_i)}{\prod_{j=1}^{M_i} e_{\pi_{ij}}\left(X_i\right)}\right]^2 \text{Var}\left(\mathbf{1}(\mathbf{W}_i = \boldsymbol{\pi}_i) \mid X_i\right)
\end{aligned}
$$

where the last equality follows from conditional exchangeability again. Next, using that treatment assignments are independent, we can further rewrite

$$
\begin{aligned}
\text{Var}\left(\mathbf{1}(\mathbf{W}_i = \boldsymbol{\pi}_i) \mid X_i\right) &= \text{Var}\left(\prod_{j=1}^{M_i} \mathbf{1}(W_{ij} = \pi_{ij}) \mid X_i\right) \\
&= \prod_{j=1}^{M_i}\left\{\text{Var}\left(\mathbf{1}(W_{ij} = \pi_{ij}) \mid X_i\right) + \mathbb{E}\left[\mathbf{1}(W_{ij} = \pi_{ij}) \mid X_i\right]^2\right\} - \prod_{j=1}^{M_i} \mathbb{E}\left[\mathbf{1}(W_{ij} = \pi_{ij}) \mid X_i\right]^2 \\
&= \prod_{j=1}^{M_i}\left\{e_1(X_i)e_0(X_i) + e_{\pi_{ij}}(X_i)^2\right\} - \prod_{j=1}^{M_i} e_{\pi_{ij}}(X_i)^2
\end{aligned}
$$

Plugging the above expression back into $(a)$, we get

$$
\begin{aligned}
(a) &= \left[\frac{\tilde{Y}_i(\boldsymbol{\pi}_i)}{\prod_{j=1}^{M_i} e_{\pi_{ij}}\left(X_i\right)}\right]^2\left[\prod_{j=1}^{M_i}\left\{e_1(X_i)e_0(X_i) + e_{\pi_{ij}}(X_i)^2\right\} - \prod_{j=1}^{M_i} e_{\pi_{ij}}(X_i)^2\right] \\
&= \left[\tilde{Y}_i(\boldsymbol{\pi}_i)\right]^2\left[\prod_{j=1}^{M_i}\left\{\frac{e_1(X_i)e_0(X_i)}{e_{\pi_{ij}}(X_i)^2} + 1\right\} - 1\right]
\end{aligned}
$$

At last, plugging our expression of $(a)$ and $(b)$ back into where we started, we obtain the final expression for the variance of the standard IPW estimator,

$$\text{Var}\left(\widehat{\phi}^{\text{IPW}}(\pi)\right) = \frac{1}{N^2} \sum_{i=1}^{M_i} \left\{ \mathbb{E}\left[ \left[\tilde{Y}_i(\boldsymbol{\pi}_i)\right]^2 \left[ \prod_{j=1}^{M_i} \left( \frac{e_1(X_i)e_0(X_i)}{e_{\pi_{ij}}(X_i)^2} + 1 \right) - 1 \right] \right] + \text{Var}\left(\tilde{Y}_i(\boldsymbol{\pi}_i)\right) \right\}$$

$\square$

## A.3 Expressing $q_{ij}$ in terms of the propensity score

We have

$$q_{ij}(\boldsymbol{\pi}_i, X_i) = \Pr(W_{ij} = \pi_{ij}, \overline{W}_i = \overline{\pi}_i \mid X_i)$$
$$= \Pr(W_{ij} = \pi_{ij} \mid \overline{W}_i = \overline{\pi}_i X_i) \Pr(\overline{W}_i = \overline{\pi}_i \mid X_i) .$$

As the propensity score $e_w(X_i) = \Pr(W_{ij} = w \mid X_i)$ is the same for every $j = 1, \ldots, M_i$, we have that all units in a cluster have the same probability of being treated. Therefore, once conditioning on the fraction $\overline{W}_i$ of treated in a cluster, the fraction equals to the probability that a unit has been treated in that cluster. We can thus write

$$\Pr(W_{ij} = \pi_{ij} \mid \overline{W}_i = \overline{\pi}_i X_i) = \pi_{ij} \cdot \overline{\pi}_i + (1 - \pi_{ij}) \cdot (1 - \overline{\pi}_i) .$$

Next, for the second probability $\Pr(\overline{W}_i = \overline{\pi}_i \mid X_i)$, we note that $\overline{W}_i = M_i^{-1} \sum_{j=1}^{M_i} W_{ij}$ can be seen a Binomial random variables scaled by $M_i^{-1}$. This means $M_i^{-1} \sum_{j=1}^{M_i} W_{ij} \sim \text{B}(M_i, e_1(X_i))$ and thus we have

$$\Pr(\overline{W}_i = \overline{\pi}_i \mid X_i) = \binom{M_i}{\overline{\pi}_i \cdot M_i} [e_1(X_i)]^{\overline{\pi}_i \cdot M_i} [1 - e_1(X_i)]^{(1-\overline{\pi}_i) \cdot M_i} .$$

Combining both expressions from above, we get

$$q_{ij}(\pi_i, X_i) = [\pi_{ij} \cdot \overline{\pi}_i + (1 - \pi_{ij}) \cdot (1 - \overline{\pi}_i)] \times \binom{M_i}{\overline{\pi}_i \cdot M_i} [e_1(X_i)]^{\overline{\pi}_i \cdot M_i} [e_0(X_i)]^{(1-\overline{\pi}_i) \cdot M_i} .$$

### A.4 Proof of Theorem 2

*Proof.* We can show that

$$
\mathbb{E}\left[\widehat{\phi}^{\text{fracIPW}}\right] = \mathbb{E}\left[\frac{1}{N}\sum_{i=1}^{N}\sum_{j=1}^{M_i}\frac{1(W_{ij}=\pi_{ij},\overline{W}_i=\overline{\pi}_i)}{q_{ij}(\boldsymbol{\pi}_i,X_i)}Y_{ij}\right]
$$

$$
= \frac{1}{N}\sum_{i=1}^{N}\mathbb{E}\left[\mathbb{E}\left[\sum_{j=1}^{M_i}\frac{1(W_{ij}=\pi_{ij},\overline{W}_i=\overline{\pi}_i)}{q_{ij}(\boldsymbol{\pi}_i,X_i)}Y_{ij}\mid X_i\right]\right]
$$

$$
= \frac{1}{N}\sum_{i=1}^{N}\mathbb{E}\left[\mathbb{E}\left[\sum_{j=1}^{M_i}Y_{ij}\mid d_{ij}(\mathbf{W}_i)=[\pi_{ij},\overline{\pi}_i],X_i\right]\right]
$$

$$
= \frac{1}{N}\sum_{i=1}^{N}\mathbb{E}\left[\mathbb{E}\left[\sum_{j=1}^{M_i}Y_{ij}(\mathbf{W}_i)\mid d_{ij}(\mathbf{W}_i)=[\pi_{ij},\overline{\pi}_i],X_i\right]\right]
$$

$$
= \frac{1}{N}\sum_{i=1}^{N}\mathbb{E}\left[\mathbb{E}\left[\sum_{j=1}^{M_i}Y_{ij}(\mathbf{W}_i)\mid X_i\right]\right]
$$

$$
= \frac{1}{N}\sum_{i=1}^{N}\mathbb{E}\left[\sum_{j=1}^{M_i}Y_{ij}(\mathbf{W}_i)\right]
$$

$$
= \frac{1}{N}\sum_{i=1}^{N}V(\pi) = V(\pi)
$$

where the second equality follows from linearity of expectations and law of iterated expectations, the fourth equality follows consistency in Assumption 1 and that the exposure mapping fulfills Assumption 2, and finally the fifth equality from conditional exchangeability in Assumption 1 because $Y_{ij}(\mathbf{w}) \perp\!\!\!\perp \mathbf{W_i} \mid X_i \Rightarrow Y_{ij}(\mathbf{w}) \perp\!\!\!\perp d(\mathbf{W_i}) \mid X_i$ for all $\mathbf{w} \in \{0,1\}^{M_i}$. $\qquad\square$

### A.5 Proof of Lemma 2

*Proof.* We have defined $\widehat{\phi}^{\text{addIPW}}(\pi) = \widehat{\phi}^{\,\beta\text{-IPW}}(\pi;\beta=1)$ which has a simpler form

$$
\widehat{\phi}^{\text{addIPW}}(\pi) = \frac{1}{N}\sum_{i=1}^{N}\left\{\sum_{j=1}^{M_i}\frac{\mathbf{1}(W_{ij}=\pi_{ij})}{e_1(X_i)} - (M_i-1)\right\}\tilde{Y}_i .
$$

As clusters are independent, we can write $\text{Var}\left(\widehat{\phi}^{\text{addIPW}}\right) = \frac{1}{N^2}\sum_{i=1}^{N}\text{Var}\left(\left\{\sum_{j=1}^{M_i}\frac{\mathbf{1}(W_{ij}=\pi_{ij})}{e_1(X_i)} - (M_i-1)\right\}\tilde{Y}_i\right)$ where the variance terms inside the sum can be decomposed as

$$
\underbrace{\mathbb{E}\left[\left(\left\{\sum_{j=1}^{M_i}\frac{\mathbf{1}(W_{ij}=\pi_{ij})}{e_1(X_i)} - (M_i-1)\right\}\tilde{Y}_i\right)^2\right]}_{(a)} - \underbrace{\mathbb{E}\left[\left\{\sum_{j=1}^{M_i}\frac{\mathbf{1}(W_{ij}=\pi_{ij})}{e_1(X_i)} - (M_i-1)\right\}\tilde{Y}_i\right]^2}_{(b)} .
$$

When $\widehat{\phi}^{\text{addIPW}}$ is an unbiased estimator, we have that $(b) = V(\pi)^2$. For $(a)$, we note that the sum $\sum_{j=1}^{M_i}\frac{\mathbf{1}(W_{ij}=\pi_{ij})}{e_1(X_i)}$ is linear with respect to $M_i$. Thus, inspecting the full expression for the variance,

$$
\text{Var}\left(\widehat{\phi}^{\text{addIPW}}\right) = \frac{1}{N^2}\sum_{i=1}^{N}\left(\mathbb{E}\left[\left\{\sum_{j=1}^{M_i}\frac{\mathbf{1}(W_{ij}=\pi_{ij})}{e_1(X_i)} - (M_i-1)\right\}^2\tilde{Y}_i^2\right]\right) - V(\pi)^2 ,
$$

we can see that the variance will scale quadratically with $M_i$. $\qquad\square$

# B  Variance reduction through augmented weighted estimators

In this work, we have considered weighting estimators that require only a single nuisance model, namely the propensity score $e_w(x) = \Pr(W = w \mid X = x)$. Because the propensity score is assumed to be known in our setting, these estimators are appealing as they eliminate the need to estimate any additional nuisance models. This, in turn, helps avoid potential biases that could arise from model misspecification. However, a key limitation of weighting estimators is that they can suffer from high variance. In this appendix, we briefly discuss a technique that introduces an additional nuisance model without risking bias, while offering the potential for variance reduction if the added model is well specified. We also present experimental results showing that this approach can indeed reduce variance without introducing bias, although the magnitude of the reduction may vary depending on the setting.

We start with a weighting estimator of the form $\widehat{\phi}(\pi; \omega) = \frac{1}{N} \sum_{i=1}^{N} \omega(\boldsymbol{W}_i, \boldsymbol{\pi}_i, X_i) \tilde{Y}_i$, where $\omega$ is a pre-specified weighting function. Both the standard IPW estimator and the $\beta$-IPW estimator can be expressed in this form using different choices of weighting functions:

$$\omega^{\text{IPW}}(\boldsymbol{W}_i, \boldsymbol{\pi}_i, X_i) = \frac{\mathbf{1}\,(\boldsymbol{W}_i = \boldsymbol{\pi}_i)}{\prod_{j=1}^{M_i} e_{\pi_{ij}}(X_i)} \quad \text{and} \quad \omega^{\beta\text{-IPW}}(\boldsymbol{W}_i, \boldsymbol{\pi}_i, X_i) = \sum_{\mathcal{U} \in \mathcal{I}_i^\beta} \prod_{j \in \mathcal{U}} \left( \frac{\mathbf{1}(W_{ij} = \pi_{ij})}{e_{\pi_{ij}}(X_i)} - 1 \right) . \quad (11)$$

This class of weighting estimators does not include the fractional IPW estimator, since it performs weighting directly on unit-level outcomes, as can be seen from inspecting (9).

We shall focus on an "augmented" variant of these weighted estimators which we define as

$$\widehat{\phi}^{\text{augmented}}(\pi; \omega) = \frac{1}{N} \sum_{i=1}^{N} \omega(\boldsymbol{W}_i, \boldsymbol{\pi}_i, X_i)(\tilde{Y}_i - \widehat{g}(X_i)) + \widehat{g}(X_i) , \quad (12)$$

where $\widehat{g}(x)$ is an estimator of the cluster-level conditional expectation of the outcome given the pre-treatment covariates, $\mathbb{E}[\tilde{Y} \mid X = x]$. The cluster-level treatment information is excluded to avoid modeling the potentially high-dimensional structure of the cluster-level treatment.

The augmented weighted estimator is reminiscent of the augmented inverse probability weighting (AIPW) estimator (Robins et al., 1994), which has been used to adjust for pre-treatment covariates with the goal of reducing estimator variance while protecting against bias when the true propensity score is known (see, e.g., Cao et al. (2009); Karlsson et al. (2024)). AIPW estimators model both the treatment-dependent conditional expected outcome and treatment probability and are doubly robust in the sense that if either of the estimators for these models is correctly specified, the AIPW estimator remains consistent. In our setting, where the design is randomized and the propensity score is known, a similar robustness guarantee holds. However, unlike the AIPW estimator, our approach relies on a prognostic cluster-level outcome model, $\mathbb{E}[\tilde{Y} \mid X]$, that does not incorporate treatment information, as would typically be required in a doubly robust estimator. The form of the augmented estimator also shares similarities with prediction-powered inference estimators (Angelopoulos et al., 2023), which use an auxiliary prediction model to improve inference, although those estimators do not employ any weighting function.

We can use any flexible, data-adaptive model to learn $\widehat{g}$, and notably, $\widehat{\phi}^{\text{augmented}}(\pi; \omega)$ is robust in the sense that it remains unbiased for the policy value regardless of whether $\widehat{g}(X)$ is correctly specified, provided that the weighting function $\omega$ satisfies some conditions. Additionally, to ensure this robustness guarantee, $\widehat{g}$ must be fitted independently of the observations used in the weighting estimator. In practice, this can be achieved using a cross-fitting procedure: the data are split into two folds, $\widehat{g}$ is fitted on one fold and predictions are made on the other fold, and then the roles of the folds are swapped and the procedure is repeated.

**Theorem 4.** *Suppose the weighted estimator $\widehat{\phi}(\pi; \omega)$ is unbiased for the policy value, $\mathbb{E}[\widehat{\phi}(\pi; \omega)] = V(\pi)$, and that the weighting function satisfies $\mathbb{E}[\omega(\boldsymbol{W}, \boldsymbol{\pi}, X) \mid X] = 1$. Furthermore, assume that $\widehat{g}$ is obtained independently of the observations used to compute its predictions. Then the augmented weighting estimator $\widehat{\phi}^{augmented}(\pi; \omega)$ is unbiased for the policy value, $\mathbb{E}[\widehat{\phi}^{augmented}(\pi; \omega)] = V(\pi)$.*

*Proof.* We can show

$$\mathbb{E}\left[\widehat{\phi}^{\text{augmented}}(\pi;\omega)\right] = \mathbb{E}\left[\frac{1}{N}\sum_{i=1}^{N}\omega(\boldsymbol{W}_i,\boldsymbol{\pi}_i,X_i)(\tilde{Y}_i - \widehat{g}(X_i)) + \widehat{g}(X_i)\right]$$

$$= \mathbb{E}\left[\frac{1}{N}\sum_{i=1}^{N}\omega(\boldsymbol{W}_i,\boldsymbol{\pi}_i,X_i)\tilde{Y}_i\right] - \mathbb{E}\left[\frac{1}{N}\sum_{i=1}^{N}\Big(1 - \omega(\boldsymbol{W}_i,\boldsymbol{\pi}_i,X_i)\Big)\widehat{g}(X_i)\right]$$

$$= \mathbb{E}\left[\frac{1}{N}\sum_{i=1}^{N}\omega(\boldsymbol{W}_i,\boldsymbol{\pi}_i,X_i)\tilde{Y}_i\right] - \mathbb{E}\left[\frac{1}{N}\sum_{i=1}^{N}\mathbb{E}\Big[1 - \omega(\boldsymbol{W}_i,\boldsymbol{\pi}_i,X_i) \mid X_i\Big]\widehat{g}(X_i)\right]$$

$$= E\left[\frac{1}{N}\sum_{i=1}^{N}\omega(\boldsymbol{W}_i,\boldsymbol{\pi}_i,X_i)\tilde{Y}_i\right]$$

$$= E\left[\widehat{\phi}(\pi;\omega)\right] = V(\pi)$$

where the fourth equality follows from the assumption that $\mathbb{E}\big[\omega(\boldsymbol{W}_i,\boldsymbol{\pi}_i,X_i) \mid X_i\big] = 1$ and that $\widehat{g}$ can be taken out of the inner expectation because it is independent of the observations used to compute its predictions. The final equality follows from that the weighted estimator $\widehat{\phi}(\pi;\omega)$ is an unbiased estimator. $\qquad\square$

To illustrate how the above theorem applies to the standard IPW and $\beta$-IPW estimators, we first note that the standard IPW estimator is unbiased under Assumption 1, and that $\mathbb{E}[\omega^{\text{IPW}}(\boldsymbol{W}_i,\boldsymbol{\pi}_i,X_i) \mid X_i] = 1$ as shown in the proof of its unbiasedness in Appendix A.2. Under the additional Assumption 3 and the study design considered in this paper, the $\beta$-IPW estimator is also unbiased, and we can similarly show that $\mathbb{E}[\omega^{\beta\text{-IPW}}(\boldsymbol{W}_i,\boldsymbol{\pi}_i,X_i) \mid X_i] = 1$ holds as follows:

$$\mathbb{E}[\omega^{\beta\text{-IPW}}(\boldsymbol{W}_i,\boldsymbol{\pi}_i,X_i) \mid X_i] = \mathbb{E}\left[\sum_{\mathcal{U}\in\mathcal{I}_i^\beta}\prod_{j\in\mathcal{U}}\left(\frac{\mathbf{1}(W_{ij}=\pi_{ij})}{e_{\pi_{ij}}(X_i)} - 1\right) \mid X_i\right]$$

$$= \sum_{\mathcal{U}\in\mathcal{I}_i^\beta}\mathbb{E}\left[\prod_{j\in\mathcal{U}}\left(\frac{\mathbf{1}(W_{ij}=\pi_{ij})}{e_{\pi_{ij}}(X_i)} - 1\right) \mid X_i\right]$$

$$= 1 + \sum_{\mathcal{U}\in\mathcal{I}_i^\beta\setminus\{\emptyset\}}\mathbb{E}\left[\prod_{j\in\mathcal{U}}\left(\frac{\mathbf{1}(W_{ij}=\pi_{ij})}{e_{\pi_{ij}}(X_i)} - 1\right) \mid X_i\right]$$

$$= 1 + \sum_{\mathcal{U}\in\mathcal{I}_i^\beta\setminus\{\emptyset\}}\prod_{j\in\mathcal{U}}\underbrace{\mathbb{E}\left[\left(\frac{\mathbf{1}(W_{ij}=\pi_{ij})}{e_{\pi_{ij}}(X_i)} - 1\right) \mid X_i\right]}_{=0}$$

$$= 1$$

The third equality follows from explicitly excluding the empty set in the power set $\mathcal{I}i^\beta$ and noting that the product over the empty set is equal to one. The fifth equality follows from the fact that, conditional on $X_i$, the treatment $Wij$ of unit $(i,j)$ is independent of all other treatments $W_{ij'}$ for $j \neq j'$. This independence allows the product to be moved outside the conditional expectation.

## B.1 Experiment

**Setup** We use the same data-generating process as in the main paper, with $N = 20{,}000$ buyers (i.e., clusters) and $K = 11$ items (i.e., units) per buyer. The interference structure is determined using the exponential decay function. To estimate $\mathbb{E}[\tilde{Y} \mid X]$, we use a linear logistic regression model, as the cluster-level outcome $\tilde{Y}$ is binary. We report the bias, variance, and mean-squared error averaged over 150 repetitions.

Table 3: Comparison of non-augmented and augmented estimators. We used $N = 20.000$ number of buyers (i.e., clusters) and $K = 11$ items (i.e., units) per buyer. We report the average bias, variance and mean-squared error with standard errors (in parentheses) over 150 repetitions.

| Estimation strategy | Bias | Variance | MSE |
|---|---|---|---|
| Standard IPW | -0.003 (0.012) | 0.737 (0.088) | 0.738 (0.032) |
| Augmented Standard IPW | 0.005 (0.010) | 0.601 (0.068) | 0.603 (0.022) |
| $\beta$-IPW ($\beta = 1$) | -0.016 (0.001) | 0.005 (0.001) | 0.005 (0.000) |
| Augmented $\beta$-IPW ($\beta = 1$) | -0.016 (0.001) | 0.004 (0.001) | 0.004 (0.000) |

**Results** As shown in Table 3, the augmented methods may reduce variance compared to their non-augmented counterparts. For the standard IPW estimator, augmentation leads to a decrease in variance (from 0.737 to 0.601) and a lower mean-squared error (from 0.738 to 0.603). For the $\beta$-IPW estimator, the variance reduction is negligible, likely because its variance is already low. Overall, augmentation with an outcome model appears to improve estimator efficiency, with the largest gains observed in cases where the initial variance is high, such as with the standard IPW estimator.

## C Estimating Qini curves in settings with no interference

We assume the following statement which is equivalent to assuming no interference.

**Assumption 4.** *We assume that $Y_{ij}(\mathbf{w}) = Y_{ij}(\mathbf{w}')$ if and only if $w_{ij} = w'_{ij}$ for all $\mathbf{w}, \mathbf{w}' \in \{0,1\}^{M_i}$.*

Note that the above assumption is a special case of Assumption 2 with the exposure mapping $d_{ij}(\mathbf{W}_i) = W_{ij}$.

We consider the simplest approach in the absence of interference for estimating Qini curves between any units. Consider the estimators based on inverse probability weighting,

$$\widehat{\phi}^{\text{no-interference}}(\pi) = \frac{1}{N} \sum_{i=1}^{N} \sum_{i=1}^{M_i} \frac{\mathbf{1}(W_{ij} = \pi_{ij})}{e_{\pi_{ij}}(X_i)} Y_{ij} \quad \text{and} \quad \widehat{\psi}^{\text{no-interference}}(\pi) = \frac{1}{N} \sum_{i=1}^{N} \sum_{i=1}^{M_i} \frac{\mathbf{1}(W_{ij} = \pi_{ij})}{e_{\pi_{ij}}(X_i)} C_{ij} .$$

We can show that under Assumption 1 and 4, the above estimators are unbiased estimators for the policy value $V(\pi)$ and $C(\pi)$, respectively. Namely, we can show this with the same proof as in Appendix A.4, but replacing $d_{ij}(\mathbf{W_i}) = [W_{ij}, \overline{W}_i]$ with $d_{ij}(\mathbf{W_i}) = W_{ij}$.

## D Experimental details

### D.1 Simulating marketplace dataset

We sample the covariates and treatment as follows: For each buyer $i = 1, \ldots, N$, we sample covariates $X_i \sim \mathrm{U}([0,1]^{12})$. Then, for each item $j = 1, \ldots, M$ we sample covariates $Z_{ij} \sim \mathrm{U}([0,1]^{11})$ and we randomize the treatment assignment by sampling $W_{ij} \sim \mathrm{Bern}(0.5)$. Here, $M$ is the same for all buyers. Since we consider the uniform cost case, we need not sample cost of treatment since they are assumed to be the same each for item.

Next, to introduce heterogeneous treatment effects, we compute an item attractiveness score matrix $\mathbf{A}$ where element $A_{ij}$ relates buyer $i$'s interest in purchasing item $j$. This matrix depends on both the covariates and treatment as follows

$$A_{ij} = \delta_{ij} \cdot (A_{ij}^{(0)} + W_{ij} \cdot A_{ij}^{(1)}) , \tag{13}$$

where $A_{ij}^{(w)} = X_i^\top \Omega_w Z_{ij}$ with $\Omega_w \sim \mathrm{U}([0,1]^{12 \times 11})$ for $w \in \{0,1\}$. The variable $d_{ij} \sim \mathrm{Bern}(0.5)$ randomly masks some elements in $A_{ij}^{(1)}$ to zero; this emulates that some items will not respond at all to a treatment. Here, $A_{ij}$ typically lies in the range $[0,1]$, but if necessary we clip it to this range so that we later could interpret it as a probability.

**Simulating revenue and cost of treating items** In our simulations, we also compute the revenue and cost of items in the marketplace if they were to be treated. This allows us to evaluate treatment policies within the simulator that prioritize items based on their predicted expected profit. Specifically, we define the price and margin fraction of item $(i, j)$ as simple linear functions of its covariates $Z_{ij}$: the price is $P_{ij} = 20 + 100 \cdot Z_{ij,1}$ and the margin fraction is $M_{ij} = 0.01 + 0.05 \cdot Z_{ij,2}$, where $Z_{ij,1}$ and $Z_{ij,2}$ denote the first and second elements of $Z_{ij}$, respectively. If the treatment $W_{ij} = 1$ corresponds to applying a fixed discount fraction of $d = 0.08$, then the revenue of treating item $(i, j)$ is $R_{ij} = M_{ij} \cdot P_{ij}$ and the cost of treating item $(i, j)$ is $C_{ij} = d \cdot P_{ij}$. Consequently, the potential profit (or loss) from treating the item, conditional on it being converted, is $H_{ij} = R_{ij} - C_{ij} = (M_{ij} - d) \cdot P_{ij}$.

## D.2 Treatment policy used for evaluation

To construct a simple baseline policy for our simulation studies, we define a treatment prioritization scoring rule $S_{\text{baseline}}$ that computes the expected profit from treating item $(i, j)$, ignoring the interference. Since our focus is on evaluating policies rather than developing them, this simplification avoids the complexity of implementing a policy that accounts for interference.

We then compute the scoring rule as

$$S_{\text{baseline}}(X_i, Z_{ij}) = A_{ij}^{(1)} \cdot H_{ij}.$$

Here, $A_{ij}^{(1)}$ represents the incremental change in the probability of conversion after an item has been treated (ignoring interference), while $H_{ij}$ denotes the profit from treating item $(i, j)$. The dependence on the covariates $(X_i, Z_{ij})$ arises through both $A_{ij}^{(1)}$ and $H_{ij}$, as described in the previous subsection.

To generate treatment policies with varying performance, we introduce a parameter $\epsilon \in [0, 1]$ used to perturb the baseline scoring rule. Specifically, we sample noise from a uniform distribution, $u \sim \mathcal{U}(S_{\min}, S_{\max})$, where $S_{\min}$ and $S_{\max}$ are the minimum and maximum observed values respectively of the scoring rule $S_{\text{baseline}}(X_i, Z_{ij})$. We then define a perturbed policy as

$$S_{\text{baseline}}(X_i, Z_{ij}; \epsilon) = (1 - \epsilon) \cdot S_{\text{baseline}}(X_i, Z_{ij}) + \epsilon \cdot u \ .$$

The best-performing policy corresponds to $S_{\text{baseline}}(X_i, Z_{ij}; \epsilon = 0)$, while $S_{\text{baseline}}(X_i, Z_{ij}; \epsilon = 1)$ is equivalent to a completely random policy.

## D.3 Hardware used for experiments

All experiments have been performed on M1 Macbooks with 16GB of RAM. The runtime is highly dependent on the number of samples being simulated, but anything under 1 million samples takes not more than 10 minutes. The total runtime of our experiments, including all variants required for uncertainty estimation, took about 3 days worth of compute time.

## D.4 Details on experiment shown in Figure 1

We simulate a dataset with $N = 20.000$ buyers (i.e., clusters) and $K = 3$ items (i.e., units) per buyer with the exponential decay function. We estimate the Qini curve for the treatment prioritization rule $S_{\text{baseline}}(X_i, Z_{ij}; \epsilon = 0)$ described in Appendix D.2 using the naive estimator and the standard IPW estimator. This was repeated 150 times and we plotted the average Qini curve.

## D.5 Additional experimental results

In this section, we present additional experimental results to support the conclusions in Section 6. Figure 4 shows the variances of the estimated Qini curves as we vary either the number of buyers $N$ (i.e., clusters) or items $M$ (i.e., units). Meanwhile, Table 1 reports the average difference in AUC between the estimated Qini curves and the ground-truth curves over multiple repetitions. These results correspond to the setting with $N = 100,000$ and $M = 11$, which is the same as for the Qini curve plots shown in Figure 3 in the main paper.

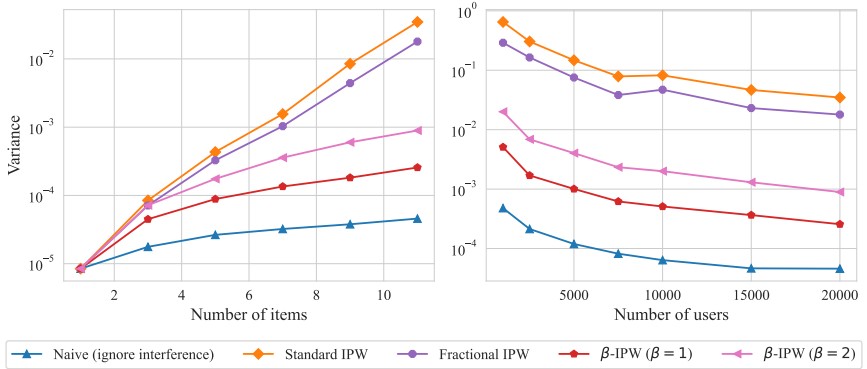

Figure 4: Comparing variance of each estimation strategy as we vary the number the buyers $N$ (i.e., clusters) or items $M$ (i.e., units) with $\eta_{\text{exp-decay}}$. While varying one, the other is kept fixed to either $N = 20.000$ or $K = 11$. The variance is reported over 150 repetitions.

Table 4: Comparison of estimation strategies via the Area Under the Curve (AUC) relative to the ground-truth Qini curve. Each entry reports the mean AUC and its standard error (in parentheses) across 150 repetitions. We let $N = 100.000$ and $M = 11$.

| Interference structure | Naive | Standard IPW | Fractional IPW | $\beta$-IPW ($\beta = 1$) | $\beta$-IPW ($\beta = 2$) |
|---|---|---|---|---|---|
| Max function | 0.11 (0.02) | 0.00 (0.05) | 0.13 (0.04) | -0.44 (0.02) | -0.28 (0.02) |
| Product function | 2.50 (0.03) | -0.04 (0.07) | 0.62 (0.06) | -0.05 (0.02) | -0.00 (0.03) |
| Exponential decay | 0.17 (0.02) | 0.01 (0.05) | 0.11 (0.04) | -0.18 (0.02) | -0.07 (0.02) |

