# OpenReview forum: "Qini Curve Estimation under Clustered Network Interference"
_TMLR — Rejected by TMLR_

### Review · Reviewer_jaZL · 2025-11-29

**Summary Of Contributions:**

The paper addresses the problem of estimating Qini curves (a widely used tool for evaluating treatment prioritization policies) when treatment interference is present. Standard Qini estimation assumes no interference between units, an assumption often violated in real-world settings such as marketing, e-commerce, and public policy, where clustered network interference is common. When interference is ignored, Qini curves become systematically biased and can mislead decision-makers about the cost-effectiveness of treatment policies.

The authors contribute the following:

- Formal Problem Setup: They define an experimental study design and identification conditions under clustered network interference, where units interfere only within independently formed clusters.

- Three Estimation Strategies: They propose and analyze three approaches for estimating Qini curves under different interference assumptions, each highlighting a bias–variance trade-off.

- Theoretical Insights: They show that stronger assumptions can produce more efficient estimators, but may introduce bias if violated.

- Marketplace Simulator: They build a simulation environment mimicking e-commerce cannibalization effects to empirically evaluate and compare the proposed methods.

Overall, the paper fills a notable gap by extending Qini curve estimation to settings where interference cannot be ignored.

**Audience:**

Yes

**Audience Explanation:**

The paper is well aligned with TMLR’s focus on machine learning methodology, causal inference, and evaluation of decision-making systems:

- Bridges a gap in uplift modeling and treatment prioritization:
Qini curves are widely used in ML applications involving individualized treatment effect estimation, uplift modeling, and personalized decision-making. Extending them to interference settings addresses a real and increasingly recognized problem.

- Addresses interference, a major topic in modern ML:
Interference is common in networked and marketplace environments, which are central to many machine learning applications. Methods that ignore interference are known to produce biased results; thus, improving robustness is directly relevant to practitioners.

**Broader Impact Concerns:**

I don't see any concerns on the ethical implications of the work.

**Claims And Evidence:**

Yes

**Claims Explanation:**

- Theoretical support:
The authors provide formal derivations and show how different assumptions affect estimator efficiency and potential bias.

- Empirical support:
The authors introduce a simulation environment that imitates real marketplace phenomena and use it to compare estimation strategies.

Therefore, the paper is well supported by both theoretical foundations and empirical demonstrations.

However, some additional results (or at least discussions) could significantly streghten the paper. In particular:

- The authors could strengthen the discussion of how their theoretical results relate to existing literature. In particular, it would be useful to address whether lower bounds exist for the variance of unbiased estimators under their assumptions, and how their findings connect to prior work on efficiency limits in the presence of interference.

- The authors should more thoroughly explain how their approach can be applied to real-world data. Although the discussion section touches on this, a deeper treatment is needed to clarify the practical feasibility of the proposed methods and the extent to which they remain valid when untestable assumptions may be violated. This would be further enhanced by including an empirical demonstration on real data, illustrating how the methods perform in practice and how sensitive they are to deviations from the theoretical assumptions.

**Requested Changes:**

Here are my main questions:

1) After Equation (3), why such R_B should be unique ?

2) How do the theoretical results relate to existing work, particularly regarding lower bounds on the variance of unbiased estimators?

Typically, for Theorem 1: Do we know whether a lower bound exists for the variance of any unbiased estimator under the assumptions stated in Theorem 1?

3) How can the proposed methods be applied to real-world data, and can the authors provide a concrete empirical example demonstrating their practical usefulness?


Minor comments:

- Equation (5): Some instances of $\pi$ appear in boldface while others do not. Please ensure consistent formatting throughout.

- Section 4.1: "Sampling variance, Since": should be a point.

- Assumption 3: The phrase “we have the an” should be corrected (likely a typographical error).

- Section 2 – Notation: Why is the first sentence—“We assume access to observations from a distribution $P$”—necessary or informative? Could you clarify the underlying space on which the distribution $P$ is defined?

---

> ### Author Response · Authors · 2026-01-08
> **Reply to review**
>
> We thank the reviewer for their questions which have helped improve our manuscript. Below you find the answer to your questions. In addition to addressing your minor comments, we have done the following revisions to our paper (see the new upload):
> - Simplified the definition we use for the Qini curve (see eq. (4)), avoiding the need to define the threshold variable $R_B$  since your comment revealed an edge case we had overlooked.
> - Introduced new experimental results using a real-world large-scale RCT from the uplift modelling literature where we artificially injected interference, effectively creating a semi-synthetic dataset with real-world characteristics.
>
> ### Clarification on $R_B$
> We thank the reviewer for this comment, which revealed an edge case we had overlooked. To address this issue, we simplify the definition of the Qini curve by removing the threshold $R_B$, which also aligns the definition more closely with the implementation in Algorithm 1.
>
> We would first like to note that under the condition that the treatment of each unit has a non-negative cost, the expected policy cost $E[C(\pi_{S,R})]$ will be a monotone function of the threshold $R$. However, treating a unit does not always result in a realized cost. For example, in an e-commerce setting where treatment corresponds to offering a discount coupon, a cost is incurred only if a conversion occurs. As a result, even though the expected policy cost is monotone in $R$, multiple thresholds may lead to the same cost level, so $R_B$ is not uniquely defined. This issue was overlooked in our original definition of $R_B$​.
>
> We therefore redefine the Qini curve $Q(R)$ as a function of the treatment threshold directly $R$ rather than indirectly via $B$. The budget is now instead defined as $B(R)$ and the Qini curve is obtained by plotting $(B(R), Q(R))$ for different values of $R$ (see eq. (4) in the revised manuscript).
>
>
> ### How do the theoretical results relate to existing work?
> Thank you for the insightful question. As we understand it, the question asks about the lower bound on the variance of any unbiased estimator under the conditions of Theorem 1, namely those in Assumption 1. The short answer is that we do not know the exact form of this bound. In principle, it is characterized by the efficient influence function of the policy value, which determines the smallest achievable asymptotic variance in a nonparametric setting.
>
> In standard causal inference settings without interference, it is well known that the IPW estimator does not have the lowest possible variance. However, the augmented inverse probability weighting (AIPW) estimator can achieve this lower bound, provided that both its nuisance components, the propensity score and the outcome regression, are correctly specified and estimated at sufficiently fast rates (Robins et al., 1994). To the best of our knowledge, comparable efficiency results have not yet been established for settings with interference. Still, it is reasonable to expect that the standard IPW estimator also does not attain the best efficiency in our setting.
>
> These observations motivated our consideration of a broader class of augmented weighted estimators, discussed in Appendix B, which are closely related to AIPW estimators. We also explain there why we do not consider AIPW estimators directly. Our results show that such augmentation can reduce variance relative to the standard IPW estimator without introducing bias, but these estimators do not outperform alternatives such as the $\beta$-IPW estimator when it comes to variance reduction. We therefore conclude that achieving meaningful variance reduction in the presence of interference may require assumptions stronger than those in Assumption 1.
>
> ### Real-world data example
> We thank the reviewer for requesting an example application of our methods in a real-world setting. As reviewer rLDp raised a similar point, and in the absence of suitable public datasets for our problem, we have performed a new experiment using  data from a large randomized experiment with no interference where we construct a semi-synthetic version by artificially inducing clustered interference. We provide further details on the dataset and the overall findings from this experiment in our response to reviewer rLDp, which we kindly invite you to read. The complete description of the experiment is also included in the revised manuscript (see the new upload).
>
> ### References
> Robins, James M., Andrea Rotnitzky, and Lue Ping Zhao. "Estimation of regression coefficients when some regressors are not always observed." Journal of the American statistical Association 89.427 (1994): 846-866.

---

### Review · Reviewer_FoVP · 2025-12-06

**Summary Of Contributions:**

The authors consider the problem of estimating a Qini curve (one of the popular evaluation scheme for treatment assignment rule) under the presence of interference (i.e., treatment on individuals can affect the outcome of others). To solve this problem, they develop an estimator of Qini curve function value by employing existing techniques for addressing interference, such as exposure map [Aronow+; 2017] and polynomial-model-based interference representation [Zhang and Imai; 2025].

### Strengths

- The problem setup seems novel, as the authors claim
- The authors derive the identifiability and the consistency of their estimators.
- The authors experimentally confirm the soundness of their estimators.

### Weaknesses

(A) The motivation of estimating Qini curve is unclear

(B) The paper suffers from clarity issues and several technically incorrect description

**Audience:**

Yes

**Audience Explanation:**

- **Novelty** and **Practicality**: To my understanding, the problem of estimating Qini curve function value under treatment interference is new. Since this problem is important for practitioners (who want to evaluate the treatment assignment rules (e.g., advertisement, personalized medicine)), I believe that some of the TMLR’s audience could be interested in how to address such a problem.

**Claims And Evidence:**

Yes

**Claims Explanation:**

- **Theoretical soundness**: Although I did not follow the proof, the authors theoretically derive the identifiability of the estimands and consistency of estimators. Establishing the technical soundness from a theoretical viewpoint is important for claiming the reliability of the results.

- **Empirical soundness**: Through simulation studies, the authors confirm the soundness of their proposed estimators.

**Requested Changes:**

Overall, the paper suffers from the clarity issues. Below I present the details.

# (A) The motivation of estimating Qini curve is unclear

The motivation of visualizing the treatment rule effectiveness using Qini curve is still unclear to me.

To my understanding, the standard Qini curve is defined using *uniform* treatment costs for all individuals and usually plots the proportion of individuals on its x-axis (not the *cost*). However, such costs, as the authors wrote, depend on the treatment and outcome of individuals, so it seems natural to think that such uniformity does not hold in practice. Due to this issue, the readers could not understand the importance of focusing on Qini curve.

The authors seem to notice this, so they consider a somewhat generalized version of Qini curve, based on non-uniform costs.

However, this context is **never** mentioned in Introduction. The following sentence is insufficient to claim the necessity of **why** we need to consider the Qini curve under interference.

> While there exists an extensive body of literature on estimating treatment effects under clustered network interference, e.g. Sobel (2006); Hudgens & Halloran (2008), little attention has been given to the problem of estimating Qini curves in this setting.

I do not still clearly understand why the authors focus on this problem and how practical the proposed evaluation scheme is.

## A-1. Relation between interference and treatment costs

Since the *costs* of treatment may depend on treatment and outcome of other individuals under interference, I believe that it is not trivial for practitioners to accurately assign the value of cost $C_{i, j}$ to each individual $j$ in a cluster $i$. For instance, if the individuals in some cluster are closely connected, treating such individuals might be risky (e.g., when taking unverified vaccin), and hence practitioners need to determine the cost value based on the evaluation of such risk.

- Please elaborate on the Qini curve more in Introduction; in particular, the meaning of x-axis in Figure 1.

- What do the authors consider as such cost assignment procedures? Please illustrate several real-world scenarios and elaborate on concrete assignment procedures; otherwise, readers cannot understand how cost $C_{i,j}$ can be given as inputs.

## A-2. Why focusing on Qini curve?

Although the authors claim that Qini curve is widely used in this field, I think that uplift curve is also commonly used. The definition of both curves seems similar, and I could not understand why the authors only focus on Qini curve through out the article, though there is no page limit in TMLR submission.

## A-3. How can we estimate the cluster assignment in practice?

Although the authors assume the access to cluster assignment of each individual, I believe that such an assumption does not always hold in practice. Elaborating on some data-driven approaches for estimating the cluster assignment is helpful for practitioners.



# (B) The paper suffers from clarity issues and several technically incorrect description

## B-1. Clarity of problem setup

- For example, the first paragraph of Section 2 is hard to follow. The authors mix the individual-based and cluster-based notation. Just consider the order: E.g., After introducing all the individual-based notation, introduce its summarized notation as cluster-based variables.

- I do not understand, but I believe that the cluster-level version of SUTVA ensures the consistency of Assumption 1, as the standard SUTVA leads to consistency assumption. How is its cluster-level version? Are there any relation to Consistency, Positibity, and Ignorability? If yes, please clearly mention the relations.

- In Section 3, they suddenly introduce the notation $R_B$, but its computation procedure is unclear from Section 3. As this notion is closely related to the x-axis of Qini curve, please elaborate on this as early as possible; otherwise, readers cannot imagine what the curve the authors want to estimate is like.

# B.2 Not exactly correct description

- For instance, in Section 3,

> A larger $S(X_i, Z_{i,j})$ should here be interpreted as that the unit (i,j) is expected to have a larger treatment effect.

seems not exactly correct. As the authors mention, the value of $S$ usually represents the CATE value, which is different from ITE (individual treatment effect). So, this should be interpreted as “units with covariates $(X_i, Z_{i, j})$”, not an individual “unit (i, j)”. I believe that the authors misunderstand what the CATE represents and the difference between CATE and ITE.

---

> ### Author Response · Authors · 2026-01-08
> **Reply to review**
>
> We thank the reviewer for the insightful comments and questions. Below, we address the concerns raised and summarize the main changes made in the revised version of the manuscript (see the new upload).
> - We now include a detailed example in the Introduction describing the application emulated in our marketplace simulator, motivating a setting in which Qini curves are useful and explaining how associated costs can be defined.
> - We have added references in the Related Work section noting that the term uplift curve is often used interchangeably with Qini curve, as well as prior work on effect estimation under unknown interference structures.
> - We added a remark following Assumption 1 discussing similarities to and differences from the traditional causal inference setting without interference.
> - We simplified the definition of the Qini curve to avoid the need to introduce the threshold variable RBRB​, thereby resolving the problematic edge case pointed out by reviewer jaZL.
>
> ### A-1: Relation between interference and treatment costs
> We shall clarify our notion of cost using the marketplace simulator from our experiments as a motivating example, which we also have added in the introduction of our revised manuscript. We have also incorporated Figure 1 into this example to help future readers understand it.
>
> In the marketplace setting, the outcome is whether a customer purchases a product, and the intervention/treatment corresponds to offering a discount. Thus, a natural way to define the cost of treatment is the financial cost of providing that discount, i.e., the discount rate multiplied by the product price. In this case, the Qini curve can be constructed by letting the x-axis represent the expected cost (e.g., the financial cost of discounts) and the y-axis the expected incremental gain (e.g., the increase in the number of purchased products), as we vary the fraction of would-be treated customers in the test dataset.
>
> While the financial cost of giving discounts can be computed in advance, it can still be viewed as counterfactual similar to the outcome, since it is only incurred if a purchase actually occurs. In other application domains, when for example considering potential side effects of a vaccine, the notion of cost may be less straightforward to quantify. Ultimately, however, defining the cost is up to the practitioner; when such (counterfactual) costs are available, our framework simply enables them to be incorporated into the Qini curve under the presence of interference.
>
> ### A-2: Why focus on Qini curves?
>
> We thank the reviewer for pointing out the ambiguity surrounding the term uplift curve, which we have found to be used inconsistently in the literature. The term Qini curve was originally introduced by Radcliffe et al. (2007), who described it as a “gains chart for uplift.” Since then, several aliases have appeared. For example, Surry and Radcliffe (2011) use the term cumulative uplift curve to refer to the same concept introduced by Radcliffe et al. (2007). The term uplift curve appears in Nassif (2013), again referring to the same curve that can be traced back to Radcliffe et al. (2007). These terms therefore describe the same underlying object.
> In this work, we adopt the term Qini curve from Radcliffe et al. (2007), as the alternative names can all be traced back to this original definition. We have included a sentence in the related work section clarifying these connections.
>
> ### A-3: How can we estimate the cluster assignment in practice?
>
> Our work assumes that clusters are naturally defined (for example, all products shown to a given customer, as in the marketplace example above). In much of the interference literature, practitioners are expected to define the interference structure based on their domain expertise. However,  in certain cases, some works have attempted to address the difficult problem of learning the interference structure from data; see, for instance, Bhattacharya et al. (2020) and Zhang et al (2023). While we do not tackle this issue in the present manuscript, we have added these references to the related work section for interested readers.

---

> > ### Author Response · Authors · 2026-01-08
> > **Reply to review (continuation)**
> >
> > ### B-1: Clarity of problem setup
> > We thank you for your suggestions. We have made several revisions to the problem setting section based on your feedback. Below, we respond to some of your remarks in more detail.
> >
> > #### Clarification on assumptions
> > In response to your questions about the Stable Unit Treatment Value Assumption (i.e., SUTVA), we would like to clarify that we do not explicitly refer to SUTVA in the problem setting section. SUTVA has two components: consistency and no interference between units. Consistency is already assumed in Assumption 1. However, SUTVA is clearly violated in our setting, since units may interfere with one another. As you point out, it is instead natural to consider an analogous SUTVA at the cluster level by assuming that clusters do not interfere with each other, which is the assumption we adopt.
> > The remaining parts of Assumption 1 (conditional exchangeability and positivity) are standard in causal inference which are unrelated to SUTVA. The main difference compared to the traditional setting without interference is that the treatment is multidimensional in our formulation of conditional exchangeability and positivity.
> >
> > To make this clearer for readers, we have added a remark after Assumption 1 explaining how our assumptions relate to, and differ from, those typically used in causal inference without interference.
> > #### Treatment threshold $R_B$
> > Regarding your remark about $R_B$, we have decided to simplify the definition of the Qini curve to eliminate the need for this threshold variable, thereby avoiding any ambiguity about how it should be computed. Another reason for this change was to resolve the problematic edge case pointed out by reviewer jaZL, we kindly refer you to read our reply to them as well.
> >
> > ### B-2: Not exactly correct description
> > We are aware of the difference between the CATE and the ITE, and we did not intend our formulation to suggest that $S$ represents the ITE in the sentence which you point out. However, based on your response, we have followed your suggestion and revised the formulation to avoid potential misinterpretation by future readers.
> >
> >
> > ### References
> > - Rohit Bhattacharya, Daniel Malinsky, and Ilya Shpitser. Causal inference under interference and network uncertainty. In Uncertainty in Artificial Intelligence, pp. 1028–1038. PMLR, 2020.
> > - Houssam Nassif, Finn Kuusisto, Elizabeth S Burnside, and Jude W Shavlik. Uplift modeling with roc: An srl case study. In ILP (late breaking papers), pp. 40–45, 2013.
> > - Nicholas Radcliffe. Using control groups to target on predicted lift: Building and assessing uplift model. Direct Marketing Analytics Journal, pp. 14–21, 2007.
> > - Patrick D Surry and Nicholas J Radcliffe. Quality measures for uplift models. submitted to KDD2011, 2011
> > - Chi Zhang, Karthika Mohan, and Judea Pearl. Causal inference under interference and model uncertainty. In Conference on Causal Learning and Reasoning, pp. 371–385. PMLR, 2023.

---

> > > ### Comment · Reviewer_FoVP · 2026-01-10
> > > **Response**
> > >
> > > I thank the authors for their reply.
> > >
> > > # (A-1)
> > >
> > > >In other application domains, when for example considering potential side effects of a vaccine, the notion of cost may be less straightforward to quantify. Ultimately, however, defining the cost is up to the practitioner; when such (counterfactual) costs are available, our framework simply enables them to be incorporated into the Qini curve under the presence of interference.
> > >
> > > Could you discuss it in Section 7 (Discussion)? It would be interesting and inspiring for future readers, as assigning counterfactual cost seems related to e.g., counterfactual harm [1]. To my understanding, this discussion is absent in the current version of the manuscript.
> > >
> > > > [1] Richens, Jonathan, Rory Beard, and Daniel H. Thompson. "Counterfactual harm." Advances in Neural Information Processing Systems 35 (2022): 36350-36365.
> > >
> > > # (A-2)
> > >
> > > I believe the Qini and uplift curves have similar definitions, but not exactly the same. For example, [2] describes the difference in Eqs. (17) and (18). But I am not sure, what do you think of this difference?
> > >
> > > > [2] Gutierrez, Pierre, and Jean-Yves Gérardy. "Causal inference and uplift modelling: A review of the literature." International conference on predictive applications and APIs. PMLR, 2017.
> > >
> > > # (A-3)
> > >
> > > I thank the authors for their revision. However, the description on cluster assignment assumption in Section 1 (Related Work) should be moved to Section 2 (Data structure & assumptions), as it is an important assumption for your proposed method.

---

> > > > ### Author Response · Authors · 2026-01-11
> > > > **Second reply to reviewer**
> > > >
> > > > Thank you again for your follow-up comments. As your suggestions could be addressed fast, we have already incorporated them into a revised version of the manuscript, which has now been uploaded. Below, we respond to each of your comments in turn.
> > > >
> > > > ## A1
> > > >
> > > > We thank you for your great suggestion, we have added a new paragraph in the discussion section discussing this topic.
> > > >
> > > > ## A2
> > > > Thank you for sharing the reference with us, it led us into doing some further investigations on this issue. Looking at eq. (17) and (18) they indeed use different definitions for the two curves. But if we inspect Figures 2 and 3 in the appendix of Gutierrez and Gérardy (2017), where the Qini and uplift curves are shown next to each other, it appears that the two curves are almost identical except for the scaling of the y-axis. For their case, it would mean that these two curves might actually not lead to different decisions if we want to do model selection based on e.g. the AUC of those curves. But we agree that there does seem to be ambiguity in the literature regarding the use of these terms. As pointed out by Devriendt et al. (2020), there is “no unique definition” of either the Qini or the uplift curve, which motivated their proposal of a taxonomy to clarify these distinctions.
> > > >
> > > > The setting we consider in our work falls into the joint-relative category in Table 2 of Devriendt et al. (2020), where the definitions of the Qini curve and uplift curve actually do coincide. By contrast, the setting considered by Gutierrez and Gérardy (2017) appears to correspond to the joint-absolute category, where the Qini and uplift curves are defined differently. More generally, differences between these definitions seem to come from different normalization choices (e.g., how to take into account the size of the treated and control group).
> > > >
> > > > For this reason, we expect that our definition can be extended to the uplift curve (when the definition differs). While it is therefore still appropriate to refer to the curve we use as a Qini curve, we have expanded the related-work section to discuss this ambiguity and to clarify how our definition fits within the taxonomy of Devriendt et al. (2020).
> > > >
> > > > ## A3
> > > > We agree with your suggestion, and have added a clarifying comment when introducing the clustered network interference in the problem setting section that the cluster structure must be known.
> > > >
> > > > ## References
> > > > - Floris Devriendt, Jente Van Belle, Tias Guns, and Wouter Verbeke. Learning to rank for uplift modeling. IEEE Transactions on Knowledge and Data Engineering, 34(10):4888–4904, 2020.
> > > > - Pierre Gutierrez and Jean-Yves Gerardy. Causal inference and uplift modelling: A review of the literature. In Claire Hardgrove, Louis Dorard, Keiran Thompson, and Florian Douetteau, editors, Proceedings of The 3rd International Conference on Predictive Applications and APIs, volume 67 of Proceedings of Machine Learning Research, pages 1–13, Microsoft NERD, Boston, USA, 11–12 Oct 2017. PMLR.

---

> > > > > ### Comment · Reviewer_FoVP · 2026-01-13
> > > > > **Response**
> > > > >
> > > > > I thank the authors for their reply and revision. Honestly, I am a bit doubtful about their expertise in distinguishing between uplift and Qini curves or in causal effect estimation. Its distinction is crucial and closely related to the article title, so I wish the authors had sufficiently reviewed and discussed it by referring to further literature.
> > > > >
> > > > > In addition, I hope the authors take my comments seriously. In particular,
> > > > >
> > > > > - I posed a question on why the authors do not consider uplift curves by illustrating their definition with a reference. However, I feel that my question remains unanswered. They just say that it is an issue of terminology. Please seriously (i) illustrate the problem setup difference (between this submission and existing work) from a viewpoint of sample size imbalance between treated and untreated individuals, (ii) discuss whether the proposed method can be extended to address the uplift curve definition in the reference.
> > > > >
> > > > > - All their revisions seem very unnatural and make the submission harder to follow. By forcefully adding a paragraph-level revision to the submission, most of the logical flows in each section are disrupted. They do not seem to be careful about readability.
> > > > > - For example, I said the assumption description should be moved to the assumption section. However, the authors only add a very short part of a sentence there
> > > > >
> > > > > > with the cluster structure known a priori.
> > > > >
> > > > > Do the authors really think this is a formal assumption statement? How is the cluster structure defined? What do you mean by "known a priori"? Totally unclear.
> > > > >
> > > > >
> > > > > Since my major concerns are clarity issues, I am unsure whether they are important in the decision-making process for TMLR acceptance. So I will make the final decision based on the discussion with other reviewers and AE.

---

> > > > > > ### Author Response · Authors · 2026-01-13
> > > > > > **Third reply to reviewer**
> > > > > >
> > > > > > We thank you for your final response and we would like to emphasize that we take your comments seriously. We also did not intend to suggest that the difference between the Qini curve and the uplift curve is just an issue of terminology. While we will not make any further revisions for the remainder of the rebuttal process, we will address each of your comments in the final version of this paper.

---

### Review · Reviewer_rLDp · 2025-12-31

**Summary Of Contributions:**

This paper addresses the estimation problem of Qini curves for evaluating treatment prioritization policies under budget constraints in settings with clustered network interference. The authors: (i) formulate policy value and policy cost under interference and define Qini curves that account for interference, (ii) demonstrate the identifiability of treatment effects under experimental designs assuming unit-independent random assignment and known propensity scores, and (iii) propose and compare three weighted estimation strategies with differing assumptions and bias-variance tradeoffs: standard cluster-level IPW, fractional exposure mapping (fractional IPW), and β-additive IPW (including β=1 and β=2). Furthermore, using a marketplace simulator with cannibalization, they compared estimator performance in terms of calibration with bias and MSE, and discrimination with policy ranking via AUC/Kendall correlation.

**Audience:**

Yes

**Audience Explanation:**

Interference arises in many areas, including marketplaces, advertising, social media, and public policy. The Qini curve is commonly used to evaluate targeting under budget constraints. This paper clarifies that ignoring interference causes the Qini curve to be systematically biased and organizes and compares estimators under interference, considering the trade-offs between strength and efficiency in the context of causal inference.

**Broader Impact Concerns:**

This paper's contribution is methodological and does not directly result in any ethical risks.

**Claims And Evidence:**

No

**Claims Explanation:**

The theoretical claims are generally clear within their scope, which includes independent unit-level randomization within clusters, known propensity scores, and independent clusters. The insight that the variance of standard IPW scales with cluster size is a convincing motivation for introducing alternative estimators. Furthermore, synthetic experiments provide a controlled comparison of interference structures and evaluate interference structures where assumptions 2 and 3 are violated.

However, their evidence is limited to synthetic experiments using simulators, and their main experiments are simplified to the uniform cost case. This makes their claim that their approach "guides the selection of the most appropriate approach” weak.  Specifically: (i) The propensity score is fixed and known. Sensitivity to estimation and estimation errors of propensities has not been verified. (ii) Exposure mapping assumes user-specified values and neglects selection uncertainty. (iii) While the general definition includes non-uniform costs, the authors only addressed the case of uniform costs. Consequently, the strongest claims are primarily supported in RCT or known-propensity settings.

**Requested Changes:**

1. Could you clarify the scope and revise the novelty positioning?
Assumption 1 and Lemma 1 are standard, and the unbiasedness component of Theorem 1 essentially follows from Lemma 1 and standard IPW arguments. The paper should clarify its novelty.

2. Could you explicitly state that the main methodology is tailored to randomized designs or known propensity scores?
Currently, the paper relies heavily on known propensity scores and uses them to avoid nuisance estimation. This must be highlighted as a core scope condition rather than an implementation detail.

3. Could you include a sensitivity analysis for propensity misspecification/estimation error?
In practice, propensity scores often need to be estimated. Since the proposed weights can involve products over cluster units, small errors in propensities can plausibly amplify with cluster size. At a minimum, conduct simulation experiments in which the analyst uses estimated or misspecified propensities, and report how calibration and discrimination metrics degrade across estimators.

4. Could you include a small real-data case study, or at least a realistic semi-synthetic one?
Even a limited demonstration using real data where naive versus interference-aware Qini curves lead to different conclusions would significantly improve external validity.

---

> ### Author Response · Authors · 2026-01-08
> **Reply to review**
>
> We thank the reviewer for their questions which have helped improve our manuscript. We address your questions below. In addition, we have done the following revisions to our paper (see the new upload):
> - Clarified the scope and contribution in the abstract and introduction, particularly that we focus on the common setting of having data from randomized experiments with known propensity scores.
> - Introduced new experimental results using a real-world large-scale RCT from the uplift modelling literature where we artificially injected interference, effectively creating a semi-synthetic dataset with real-world characteristics.
>
> ### Scope and novelty positioning
> We thank the reviewer for the question.
>
> The  scope of our research in this manuscript was to develop and study estimation strategies for estimating Qini curves under clustered network interference in the setting with data from a randomized experiment. This setting closely resembles the standard setting in e.g. the uplift literatures where Qini curves are commonly used, with the non-trivial novel addition that we allow for the presence interference compared to previous work.
>
> Among our theoretical results, some have indeed appeared in various forms in e.g. the non-interference literature, such as the identification result in Lemma 1 and unbiasedness in Theorem 1. Meanwhile, the variance arguments we make in Theorem 1, the paragraph under Theorem 2, and the results from Lemma 1 are novel and critical in our understanding of the estimators’ behaviour in our experiments. We did additional investigations such as those in Appendix B which we did not include in the main paper, but which further complement our theoretical results.
>
> We have clarified these contributions more in the introduction section.
>
> ### Uniform cost case in experiments
> We agree that it may be confusing that we introduce the non-uniform cost setting earlier in the paper but focus on the uniform cost case in our experiments. As stated at the beginning of Section 4, however, the estimation strategy is the same in both cases; only the identification assumptions need to be stronger in the non-uniform cost setting. Consequently, we can study the estimation strategies in the uniform cost case without loss of generality. The findings discussed in our Discussion section, which primarily concern comparisons between different estimation strategies, thus apply equally to both the uniform and non-uniform cost cases.
>
> To avoid confusion, in addition to the existing remark at the start of Section 4, we have now added another clarifying remark to Section 3, where the uniform cost setting is first introduced.
>
> ### Specify that method is meant for experimental data with known RCT
> We thank the reviewer for noting that the scope of our problem setting was not fully specified in the introduction section. While the study design is described in detail in Section 2, we have now also clarified in the abstract and introduction that our method is intended for settings in which data comes from randomized experiments and, consequently, the propensity score is known.
>
> ### Misspecified propensity score
>
> We would like to push back on the premise that “in practice we always estimate the propensity score.” In the original marketing application considered by Radcliffe (2007), treatment is randomized between control and treatment groups, so the propensity score is known by design. Similarly, for instance in medicine, randomized controlled trials are the standard for obtaining regulatory approval of new drugs, and in such settings the propensity score is again known a priori. There are therefore many important applications in which estimating the propensity score is unnecessary.
>
> We agree, however, that it is also important to consider settings in which the propensity score must be estimated, such as observational studies with non-randomized data. While we believe that our estimators can be extended to this case, we believe studying the impact of estimation error and potential misspecification on the resulting Qini curves is beyond the scope of the present paper.

---

> > ### Author Response · Authors · 2026-01-08
> > **Reply to review (continuation)**
> >
> > ### Real-world case study
> > We thank the reviewer for emphasizing the need to evaluate our methods on a more realistic dataset in addition to the simulator. We were unable to identify any public datasets with clustered network interference that are sufficiently large to support learning heterogeneous treatment effects and evaluating personalized treatment policies. We therefore turn to a large randomized advertising experiment where no interference is present from Eustache et al. (2018) that is widely used in the uplift modeling literature, and construct a semi-synthetic version of this dataset by injecting artificial clustered interference while preserving the original data distribution as much as possible. The dataset was collected over a two-week period and contains approximately 14 million users with 12 covariates, a binary treatment indicating whether an advertisement was shown, and a binary outcome indicating conversion on the advertiser’s website.
> >
> > In this experiment, we compare two policies derived from different CATE estimators, where one policy is intentionally degraded by adding a small amount of noise to its predictions. Consistent with our simulation results, we find that the naive estimator exhibits substantial bias, whereas all proposed methods correct for this bias to varying degrees. Notably, among our proposed estimators, only the $\beta$-IPW estimator is able to reliably discriminate between the two policies, as it achieves the largest variance reduction and therefore produces non-overlapping confidence intervals for the AUC. These findings are all consistent with those from our simulation studies.
> >
> > We provide full details of the data construction and complete results in the revised manuscript.
> >
> > ### References
> > Diemert Eustache, Betlei Artem, Christophe Renaudin, and Amini Massih-Reza. A large scale benchmark for uplift modeling. In Proceedings of the AdKDD and TargetAd Workshop, KDD, London,United Kingdom, August, 20, 2018. ACM, 2018.

---

### Decision · Action_Editor_vomQ · 2026-02-11

**Recommendation:** Reject

**Additional Comments:**

The reviewers agree that this study is well-motivated.
However, they pointed out some issues in the writing, and these issues are closely related to the paper’s claims of novelty.
While it is understandable that the authors wish to highlight the significance of their work, the current manuscript appears to overstate its contributions.
For publication, the authors should narrow the scope of their claims.

I would suggest the authors to look into the acceptance criteria of TMLR.
https://jmlr.org/tmlr/acceptance-criteria.html

The reviewers’ major concerns are as follows.

> The paper treats the Qini curve and the uplift curve as equivalent, which may lead to an overstatement of the generality of the results.

In the revision, the authors cite Devriendt et al. (2020) and state that “the Qini curve and uplift curve happen to coincide.”
While they do coincide for the joint-relative curves in Table 2 of Devriendt et al. (2020), this equivalence does not hold in general.
Based on this single observation, the statement that the authors “expect our framework to extend naturally to the other categories in their taxonomy” may overclaim the generality of the results.
Moreover, this intentional conflation may be seen as showing insufficient respect for the distinct historical development of research on the Qini curve and the uplift curve.

> The paper assumes that the propensity score is known, but this assumption is generally too strong.

The authors argue that the propensity score is known in randomized experiments, which is correct.
However, whether the propensity score is known or unknown can substantially change the difficulty of the problem.
The manuscript does not sufficiently discuss this point.
Because this is a central assumption underlying the entire analysis, it should be discussed more carefully and emphasized more clearly in the paper.

**Audience:**

Yes

**Audience Explanation:**

Because the Qini curve plays an important role in causal effect estimation, some readers of TMLR will find this paper of interest.

**Claims And Evidence:**

No

**Claims Explanation:**

This paper investigates the estimation bias of the Qini curve under a clustered network setting.
The authors proposed three unbiased Qini-curve estimators under different assumptions, and evaluated their performance using synthetic data.
The experimental results suggest that the estimator relying on stronger assumptions achieves smaller variance than those based on weaker assumptions.

The reviewers agree that the study is well-motivated.
However, they identified some issues with the writing.
In particular, the major concerns are as follows:
* The paper appears to treat the Qini curve and the uplift curve as equivalent, which may lead the authors to overstate the generality of their findings.
* The analysis assumes that the propensity score is known, but this assumption is generally too strong in practice.

**Resubmission Of Major Revision:**

The authors may consider submitting a major revision at a later time.